# A Novel Camera-Based Approach to Increase the Quality, Objectivity and Efficiency of Aeronautical Meteorological Observations

**Juraj Bartok [1,2], Lukáš Ivica [1], Ladislav Gaál [1,*], Ivana Bartoková [1] and Miroslav Kelemen [3]**

[1] MicroStep-MIS, Čavojského 1, 841 04 Bratislava, Slovakia; juraj.bartok@microstep-mis.com (J.B.); lukas.ivica@microstep-mis.com (L.I.); ivana.bartokova@microstep-mis.com (I.B.)

[2] Department of Astronomy, Physics of the Earth, Meteorology, Comenius University in Bratislava, Mlynská Dolina, 842 48 Bratislava, Slovakia

[3] Faculty of Aeronautics, Technical University of Košice, Rampová 7, 041 21 Košice, Slovakia; miroslav.kelemen@tuke.sk

[*] Correspondence: ladislav.gaal@microstep-mis.com

**Abstract:** Despite the general efforts of meteorological services to provide aeronautical observations at all ranges of airports automatically, for some meteorological variables, especially for the ones that highly rely on complex human perception (e.g., prevailing visibility and cloud coverage), reliable, fully automated observations cannot be ensured. This paper introduces novel possibilities to observe prevailing visibility and cloud coverage/height by means of a camera-based observation system that does not necessarily replace, but effectively and synergically amends the standard observations. We present human (and not automated) observations from a remote center that allows for an observer to report meteorological conditions remotely, only using images from cameras installed at the airport. The basic concept of the remote observer was developed within a previous SESAR project. The focus of our methods is set (1) on the quality of information with the occurrence of reduced visibility and enhanced cloud cover in inhomogeneous weather situations and (2) on a comparison of our approaches with those from local human observers. We conclude that for a correct estimation of the prevailing visibility, cloud coverage and cloud types, the automated sensors alone are inadequate; however, the camera-aided remote human approach to observations seems to be a promising supplement to eliminate the sensors' deficiencies, in terms of the quality (e.g., high quality camera records; no more point measurements), objectivity (e.g., database of archived weather situations) and efficiency (e.g., no need to have an observer physically present at the airport). The possibility to provide observations remotely seems to be advantageous in the COVID-19 and post-COVID-19 era when the society must adapt to different levels of quarantine conditions, affecting and/or disabling standard work and travelling regimes.

**Keywords:** camera-based remote observer; aeronautical meteorological observations; visibility; clouds; cloud coverage; cloud base height

## 1. Introduction

Needs for efficiency in airport operations and quality requirements put pressure on the performance of the airport Automated Weather Observation System (AWOS) at all ranges of airports from the small to the largest ones. Nevertheless, 24/7 human observations carried out locally at small or medium airports may not be sustainable economically: providing the necessary local staff to operate such an airport can be challenging and costly. Therefore, observations of the current weather by professional observers in these locations are often completely or partially replaced by simplified ones. Replacement by the way of reporting from a human-based to a standard, automated, and point sensor-based one, especially when reporting prevailing visibility and cloud coverage, however, may have quality issues

and may cause risks mainly in case of hazardous weather phenomena in the airport vicinity or reduced visibility at critical distances (Figure 1).

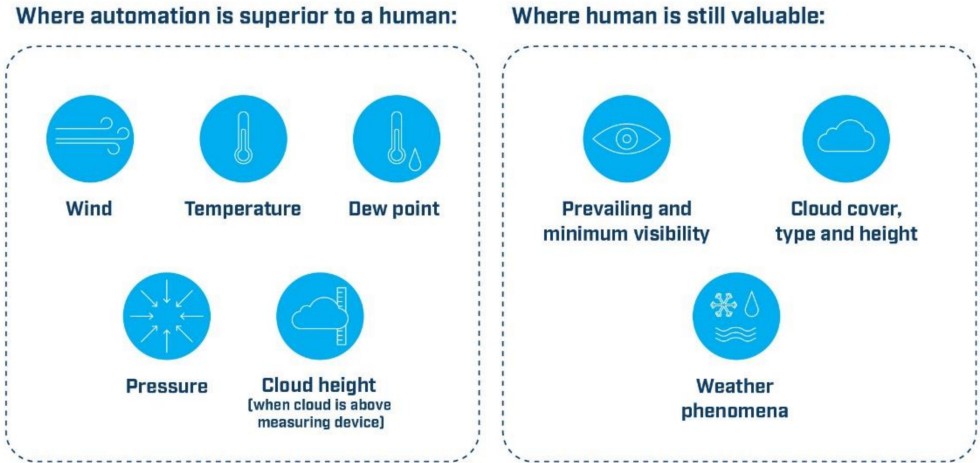

**Figure 1.** Categorization of the basic meteorological variables to those easily replaceable by automatization (**left panel**) and to those where human observers are still highly valuable (**right panel**).

The motivation for researching the topic of the paper is also its perception as a parallel agenda with the concept of 'Remote and Digital Towers'. The concept of the 'digital tower' was introduced to capture the growing demand for the integration of a wide range of air traffic management systems and data to support advanced air transport operations (SESAR—Single European Sky Advanced Research Joint Undertaking program etc.). In this way, aviation practice also creates space for innovative approaches to the provision of digital aeronautical meteorology services. The key benefits of the proposed solutions for remote sensing and data evaluation will also be reflected in this new workspace, with an emphasis on improving professional information provision with meteorological data for flight crews and air traffic controllers, to maintain a high standard of safety for both air traffic and flight management. Remote access may enable more efficient use of available human resources, improve employee workload management, and improve work organization up to the so-called home office for selected professions that are and remain the challenges in the current pandemic period, but also for the post-COVID-19 period. The proposed solution has the potential to improve the decision-making processes of meteorologists and to reduce the scope for human error and subjective data evaluation. Finally, it may allow for a more efficient use of funds for the provision of this service and for professional staff, but also for the required infrastructure. We expect to see application examples of the impacts of innovative digital approaches in the reduction of the number of real buildings for air traffic control and in the reduction of finances for their operation, maintenance, and modernization, etc. Such application impacts can also be expected in connection with the infrastructure for the provision of aeronautical meteorological service at small, regional airports, etc., where remote access will be implemented. The information can be shared not only by air transport but also for the benefit of general aviation, thus expanding the application area of remote access users of the aeronautical meteorological service.

In this paper, the focus is set on a novelty approach to visibility assessment and cloud observations, for several reasons. Good visibility, both in horizontal and vertical directions, is a crucial element in the safety of all kinds of transport (road, sea, or air) as low visibility may result in fatal accidents. It is especially important in aviation when the aircraft is maneuvering on or near the ground. Similar considerations are also valid for significant cloud layers, usually below 1500 m (5000 ft), or below minimum sector height, whichever is greater [1]. Nevertheless, both visibility and cloud observations, regardless of whether they are observed at large airports or smaller ones, are associated with several weaknesses, e.g.:

- Subjective character of observations. Cloud coverage and visibility are estimated by human observers, and, thus, the objectivity of the observations cannot be guaranteed. Even two professional and experienced observers may evaluate a given meteorological situation in a slightly different manner.
- Immediate nature of the phenomena. Cloud coverage and visibility are meteorological variables that are estimated instantaneously, and no visual records of the current meteorological conditions are taken. This may, however, be a problem, for instance in the case of an investigation of the meteorological circumstances of an aircraft accident.
- No obstacle-free view. It happens often that the meteorological observatory is surrounded by buildings, trees, or other natural or man-made objects that limit the obstacle-free view of the observers in all the possible directions.

Automated sensors are able to overcome some of the aforementioned issues; however, they also have their own deficiencies, as will be discussed below.

Cloud coverage has long been estimated only by human observers. The advantage of such a traditional approach is that a human observer can perceive the whole sky, and thus, is able to estimate the cloud coverage. Conversely, the most significant drawback of this method stems in the degree of the related subjectivity, as mentioned above. Beyond this, there are further limitations of human observations. First, it is difficult to carry out precise observations during the night, and secondly, there are practically no capacities to ensure continuous human observations.

The development of measuring equipment, mostly of those based on laser technology, then brought opportunities towards automatization of cloud observations. Recently, laser ceilometers are used to estimate the heights of different cloud layers. Ceilometers emit short laser signals towards the sky, and determine the height by measuring the time it takes for a pulse of laser light to be scattered back from a cloud base (or ceiling). The advantage of the ceilometers lies exactly in the high precision of their measurements of cloud height as well as in their ability to work in a continuous regime. However, the largest disadvantage of the ceilometers is the fact that they only report point measurements above the ceilometer location (or, at larger airports, there are a few points available). Consequently, the point character of the ceilometer measurements is the main reason why they are not suitable for a correct estimation of the cloud coverage. Ultimately, cloud type cannot be determined by ceilometers at all.

Automated tools also do exist to measure the second meteorological variable discussed herein, i.e., the visibility. There are generally two types of sensors to determine prevailing visibility: transmissiometers and forward scatter sensors. These operate on the basis of different principles. Transmissiometers measure the transmittance, i.e., a beam of light is emitted horizontally to a sensitive receiver several meters away, and the visibility is determined proportionally to the reduction in the signal intensity. In contrast to this, forward scatter sensors make use the principle of scattering by particles that are suspended in the air. In this type of sensor, a beam of light is emitted at an angle from a sensitive transmitter, and the amount of light scattered into the receiver is measured. By choosing the correct angle between the transmitter and the receiver, the extinction coefficient is estimated. Forward scatter sensors are suitable at measuring the transparency of the atmosphere; nevertheless, they only supply a short distance measurement of air between the transmitting and receiving heads (on the order of a few tens of cm, according to the manufacturer).

Automated visibility sensors are generally designed to measure the prevailing visibility by assuming that the conditions between the sensor's receiver and transmitter represent the nominal conditions around the horizon. As the actual visibility may not be homogeneous over the entire domain, it is quite possible that the visibility estimate of the laser sensor could differ from that of a human observer [2]. That is why differences between forward scatter sensors, transmissiometers, and observers exist.

Transmissiometers and forward scatter sensors are associated with a number of further drawbacks: (a) they are scarcely available uniformly, and their spatial distribution is far from ideal, even within a given country, (b) they can only measure visibility at exact, pre-defined measuring points (Figure 2), (c) their price is relatively high, and (d) they are unable to spot minimum visibility and its direction that is required by the International Civil Aviation Organization (ICAO) [1].

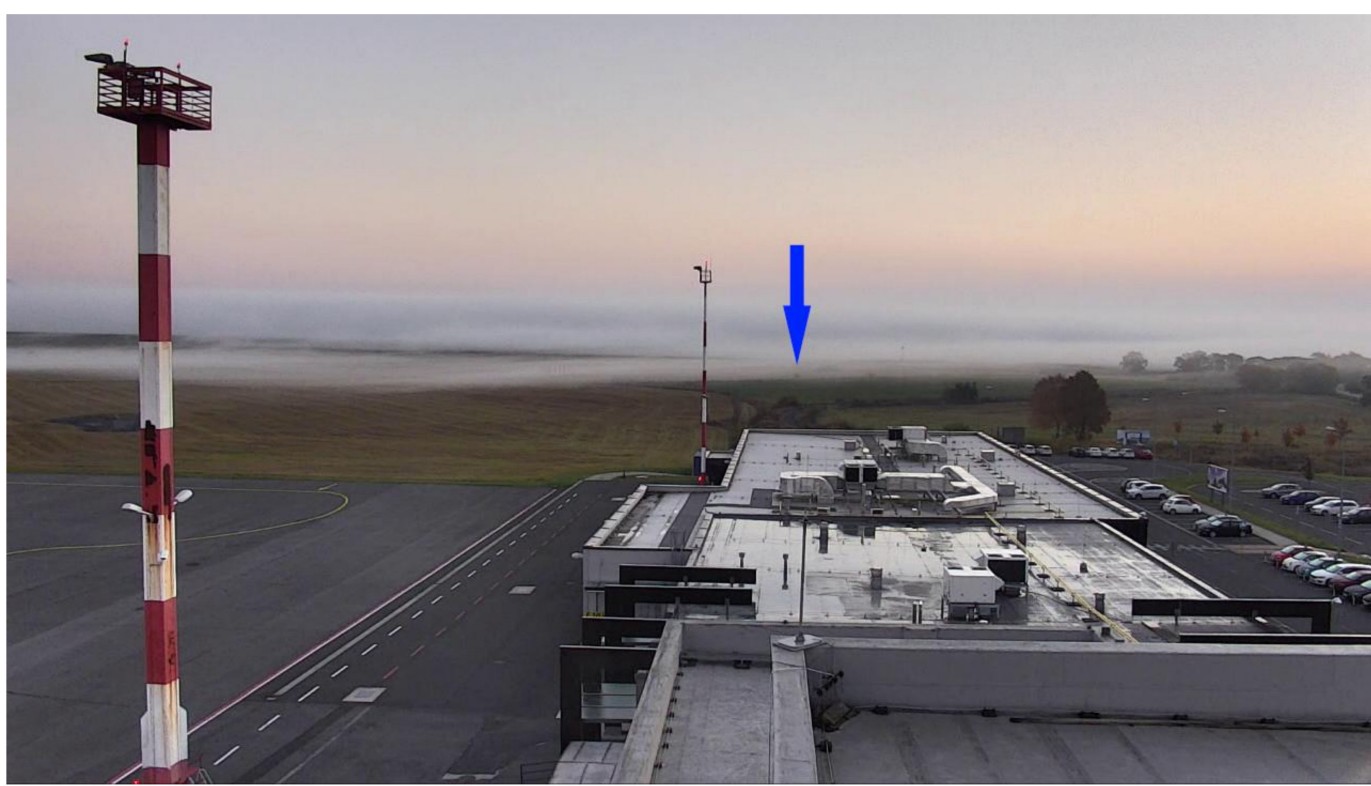

**Figure 2.** An example to illustrate the drawbacks of point measurements of visibility. Here, the visibility meter (indicated by the blue arrow) gets hidden in a fog patch and reports decreased visibility, but the overall prevailing visibility is higher.

A novel approach to a determination of visibility by remote and/or automatic sensors is using camera images. This solution addresses all the above-mentioned problems (a)–(d) as: (a) a lot more cameras than sensors operate on a regular basis, but they are generally not used to estimate visibility; (b) cameras can cover the entire environment in contrast to the visibility sensors; (c) the price of cameras is lower than that of the dedicated, high-quality sensors; and (d) as a result of a full coverage of the whole environment, the minimum visibility and its direction can be determined. Generally, the camera-based approach to the visibility measurement is rare and novel, but not unseen [3–5]. The research and development of such systems is more advanced in the road visibility domain [6,7].

A similar, camera-based approach is also aimed at getting over the weaknesses of cloud observations. Its basic idea is to take a photo of the full hemisphere [8], and carry out an expert analysis of each picture to determine the cloud coverage. Furthermore, the base height of the cloud cover can also be estimated, on the basis of (i) ceilometer data, (ii) camera images of visibility (e.g., cloud layers relative to the nearby terrain subjects such as mountain peaks), and (iii) photos of the full hemisphere. Alternatively, a more straightforward way to estimate the base height of the cloud cover is to take photos of the hemisphere by an infrared camera, and calibrate the observed cloud temperatures by means of the information on the vertical air temperature profile.

The camera-based approach to observation of both meteorological variables is integrated in the proposed system, which will be presented later in the paper. It attempts to overcome the drawbacks of the aforementioned sensors by covering the entire surrounding environment. An essential part of the proposed system is a database of observations, which allows for retrospective analysis of weather events and their verification any time. The most important feature of the system is its remote character, i.e., the meteorological observations and the subsequent analysis of the variables may be carried out anywhere, so the observer does not have to be physically present at the target destination. The basics of the proposed system were laid down in the framework of a SESAR program. The SESAR project Pj.05 was dedicated to Remote Towers, i.e., to developing air traffic services by a remote Air Traffic Controller. Following this, the project 'Pj.05-05 Advanced Automated MET System' launched our experiments in an accompanying service, termed as Remote (meteorological) Observer.

The aim of the paper is to describe new possibilities of observing the prevailing visibility and cloud coverage/height using a camera-aided observation system, which does not necessarily replace, but effectively and synergistically complements standard observations. Note that the proposed solution is not a fully automated, computer-vision-based one. The developed system still relies on the perception, knowledge, and experiences of the human observers, though the assistance of advanced technological utilities.

The paper focuses on validation experiments to confirm our preliminary hypotheses, which expect that (1) the remote observer approach would show better performance than the standard AWOS, and, at the same time, (2) the performance of the remote observer approach would be comparable to local observations, to estimate cloud base height, cloud coverage, and prevailing visibility.

The paper is structured as follows: In the next section, the general architecture of the camera-based system for the remote observation of visibility and cloud coverage is introduced, including the collection of the camera records from the target area, their processing, labeling, backup, and their visualization within the human–machine interface of the proposed system. The third section presents the results of the validation experiment that predominantly focused on the comparison of the novel, remote approach to observations with the standard ones, obtained both by means of automated weather observation system and by the local professional observers. First, the properties of the clouds are examined (the cloud base height and the cloud coverage), which are illustrated by several characteristic case studies. A similar approach is adopted in the case of the prevailing visibility. Finally, the first experiences with the performance of the proposed system are discussed, with a focus on the new perspectives to improve the system in the near future.

## 2. Materials and Methods

### 2.1. General Architecture

The general architecture of the examined solution, which includes subsystems for the observation of both the cloud coverage/heights and the prevailing visibility, can be characterized as follows. A visual camera attached to a rotator with accessories is installed on a high point at the airport with an obstacle-free view. The camera sends images of the sky/horizon to the central server every 5 min. Note that the frequency of shooting is customizable: one can also get to a 1 min periodicity if needed.

In the case of cloud observations, the algorithm of stitching the individual photos of the sky together and its subsequent computer processing is performed, after which the whole sky image is presented to the remote observer (described in detail in Section 2.2). In the case of the visibility observations, pre-defined distance markers are added to the images. With the help of these markers, the remote observer determines the prevailing visibility and saves the results to the server (more detail in Section 2.3).

Additionally, a complete METAR (METREPORT/SPECI/SPECIAL) (METAR—Meteorological Aerodrome Report, or Meteorological Terminal Air Report, or Meteorological Terminal Aviation Routine Weather Report) message can be generated. The original and the processed images are stored in a database and are also available through a web-based human–machine interface (HMI) to a remote human observer, who can be located anywhere with Internet access. The database allows for any posterior verification of the decision procedure. The output information on the meteorological variables is then sent to the air traffic control (ATC) tower or any institution directed.

The basic workflow of the system and its components at the Poprad-Tatry Airport (ICAO: LZTT; IATA: TAT) is visualized in Figure 3. The red rectangle denotes the automated workflow, whereas the green one indicates the intervention by a (remote) human observer.

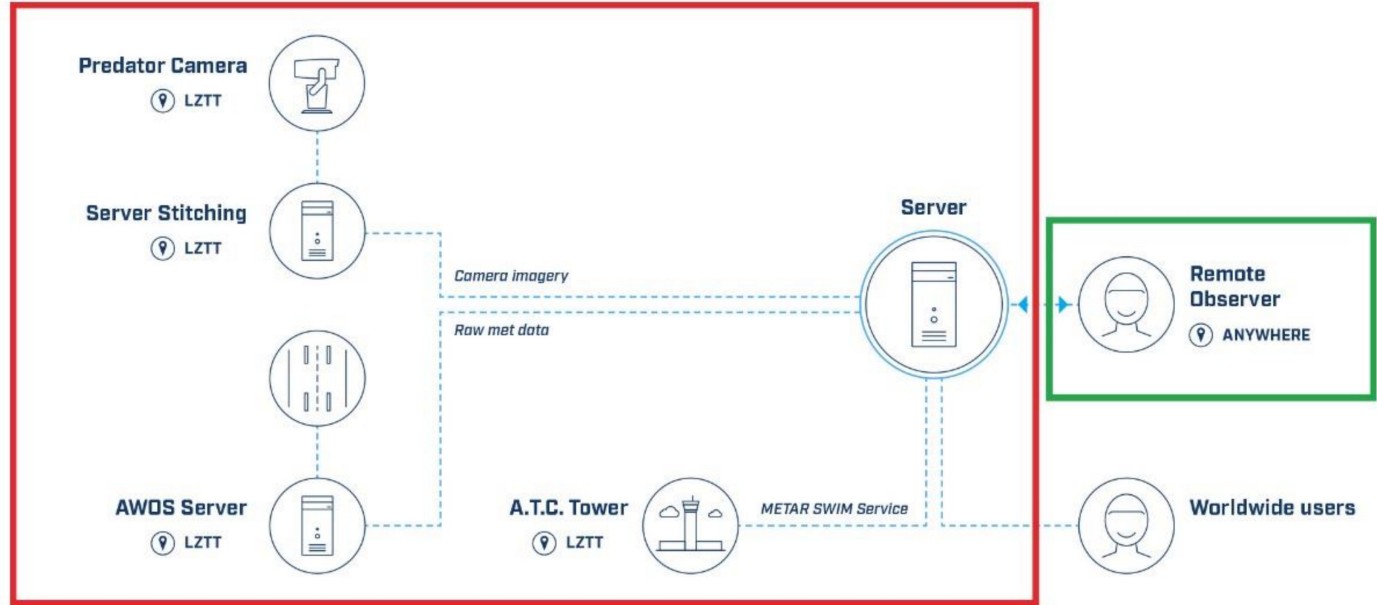

**Figure 3.** The structure of the proposed system, as designed at LZTT.

*2.2. Cloud Observations*

Before describing the novelties of our proposed system, one has to be familiar with the standard procedures of estimating cloud properties by meteorological observers. Cloud base height is currently observed at airports worldwide in three steps:

(1) The human observer looks to the sky, and recognizes different cloud layers and cloud types, which indicate initial information of their typical height.

(2) If there are some tall country features nearby, the observer utilizes them in a more precise estimation of the cloud base height. If such tall features are not available, this step is skipped.

(3) In the workroom, the observer checks the ceilometer measurements of the cloud base height, and although they are only point measurements, these are utilized in a combination with Steps #1 and #2 to get as comprehensive information on the cloud base height as possible.

The steps are not necessarily performed in the described order. The specific cases with no country features nearby (i.e., without Step #2) and/or with no ceilometer (i.e., without Step #3) will be discussed in Section 4 (Discussion).

In our system, the remote human observer can perform Steps #1 and #2 remotely by cameras and Step #3 by a remote access to the ceilometer data, which are displayed in the same system as the camera imagery. In this way, it is ensured that the remote observer has the ability to perform the same steps as the local observer.

To test the functionality of the semi-automated subsystem targeted at the observation of cloud coverage and cloud heights, two cameras were installed at LZTT. The first one is aimed at taking photographs in the visible RGB spectrum (advantageous during the daytime), whereas the second, thermal camera is used to monitor the sky in the infrared spectrum (advantageous during the nighttime).

At each observation time, both cameras take 61 photos of the sky, at a pre-defined angles and azimuths. In the first step of the processing, these images are stitched together, using the method of Lambert azimuthal equal-area projection from the hemisphere to the horizontal plane (Figure 4). The selection of an equal-area projection is important, as the ratio of the cloud surface to the whole sky surface in reality must be exactly the same as the ratio in the whole sky image [9].

**Figure 4.** A schematic of the Lambert azimuthal equal-area projection.

The procedure of accurate image stitching needs special attention. Presets for every sub-image are always the same as the camera with the rotator were well fixed on the same place for the whole duration of the research. Countermeasures against possible differences from the pre-defined preset positions have been conducted. The camera with the rotator has a proclaimed operational wind load durability 216 km/h (135 mph), and during the research operation, no considerable wind gusts were observed. The 1–2 pixel movements could possibly be caused by the returning of the camera to the same preset position from a different side in the case of power supply outage, but such a difference is negligible for the final output and its usage. Overlapping parts of the sub-images are averaged if an image is darker than the neighboring one.

Even though the procedure of image stitching is not trivial, we decided to use a rotating robotic camera instead of the commonly adopted fisheye lens approach [10–13]. While the latter methodology produces easy-to-get images, their resolution may not be sufficient for the remote observers, for instance to distinguish sharp edges of the cloud layers.

The HMI of the proposed system offers the remote observer several aiding tools to make the assessment of the cloud cover easier. These include displaying concentric circles and/or dividing the photo of the hemisphere into oktas (Figure 5). The concentric circles are useful, for instance, when clouds are close to the horizon, whereas the delineation of the hemisphere into oktas may be helpful when assessing cloud coverage that only appears in certain parts of the sky. A further option is to display the location of the ceilometer on the stitched image of the hemisphere.

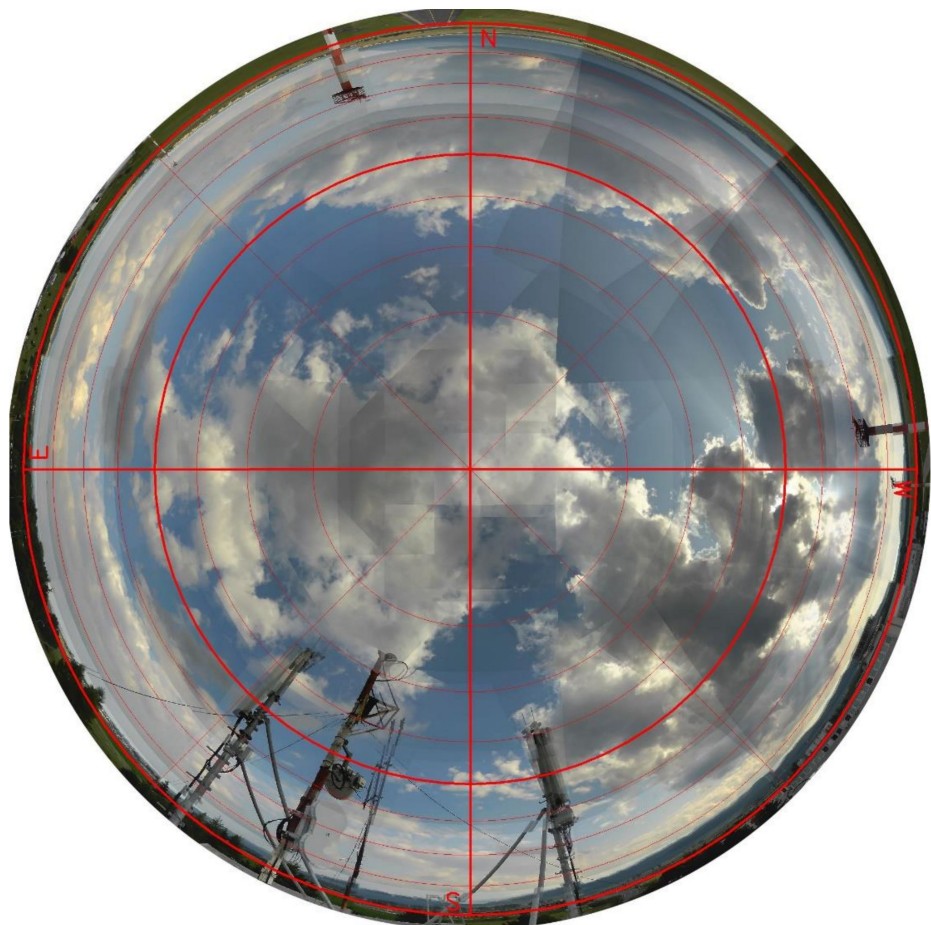

**Figure 5.** Aiding tools for the assessment of the cloud coverage in the human–machine interface of the proposed system.

Note that the switched position of east/west in Figure 5 is not a mistake. This is exactly the way the observer sees the cardinal directions when looking up to the sky with its head pointing to the north.

The procedure of processing the stitched infrared images is similar to that of the RGB's; nevertheless, the algorithm also makes use of further meteorological information (from the ceilometer if available, from high-altitude aerological balloons if available, and other weather data), to derive the actual vertical profile of the temperature, and thus, to calibrate the temperature of the infrared images.

Note that in the current study, we only present some initial analysis of the sky images from the RGB camera, and no data from the infrared camera will be examined.

### 2.3. Visibility Observations

To allow for a remote observation of visibility, a high-resolution camera for the visible spectrum was installed at LZTT, where the prevailing visibility in METAR messages is reported every 30 min by professional observers. The visible camera installed on a rotator sends 8 images of the horizon covering all the cardinal directions (N, NE, E, SE, S, SW, W, and NW) to the system. Based on these images, the proposed software provides an easy-to-use HMI and tools for the remote observation of the prevailing visibility as close to the estimation of the local observer as possible.

Markers with distance labels for all directions represent an aiding tool for the remote observer. These markers were selected carefully to cover the variability of distances in each direction. The number of markers is normally superior to the number of local observations, and thus, it makes the observation of the prevailing visibility more precise. Table 1 presents a basic overview on the marker settings at LZTT. The marker counts were divided into four intervals that are being used by the ICAO [1]. The observations evaluated by means of the proposed system are based on a multitude of markers used by local observers and this may lead to differences in statistics.

**Table 1.** The number and the coverage of the reference cardinal directions covered by markers used by local human observers (LO) and the proposed system for remote observation (RO), respectively.

|  | Observer | 0–600 m | 600–1500 m | 1500–5000 m | >5000 m | Sum |
|---|---|---|---|---|---|---|
| Number of reference markers | LO | 15 | 3 | 8 | 20 | 46 |
|  | RO | 56 | 26 | 30 | 41 | 153 |
| Number of directions covered by markers | LO | 8 | 2 | 4 | 8 | — |
|  | RO | 8 | 8 | 8 | 8 | — |

By switching markers between the statuses visible/not visible, the visibility in each direction and thus, the prevailing visibility can automatically be re-calculated. Beyond this, once the visibility markers have been set, they are available for any later verification or correction by the remote observer (Figure 6). One can easily access the current images, history of images, or images related to ideal conditions for a quick comparison to improve the observer's decision on the visibility. The system settings also support generating METAR messages, where all the fields are automatically filled in and the observer only edits unautomated fields or fields with a correct suggestion from a sensor. The prevailing visibility determined through the remote observer's HMI is also pre-filled in the METAR generation tool.

For a possible comparison of the remote observations with the METAR messages, the remote observations were timed 10 min before METAR was issued, as according to local references, this is the approximate time when local observations should take place. Generally, there is no obligation for the local professional observer to carry out observations at specific times [14]. Unfortunately, such a vagueness of the definition of measurement times may be one of the potential sources of the inconsistencies between the measurements (remote observations vs. METAR messages) and it has to be accounted for in their interpretation.

### 2.4. Data Sets and Methods

The basic evaluation of the proposed system took place between 23 November 2018, and 18 December 2018, at the Poprad-Tatry Airport (LZTT), with the remote assistance of the professional meteorological observers on duty at Bratislava Airport (LZIB). It was a real-time experiment, i.e., the professional observers at Bratislava carried out parallel observations both at LZIB and LZTT as the weather was happening.

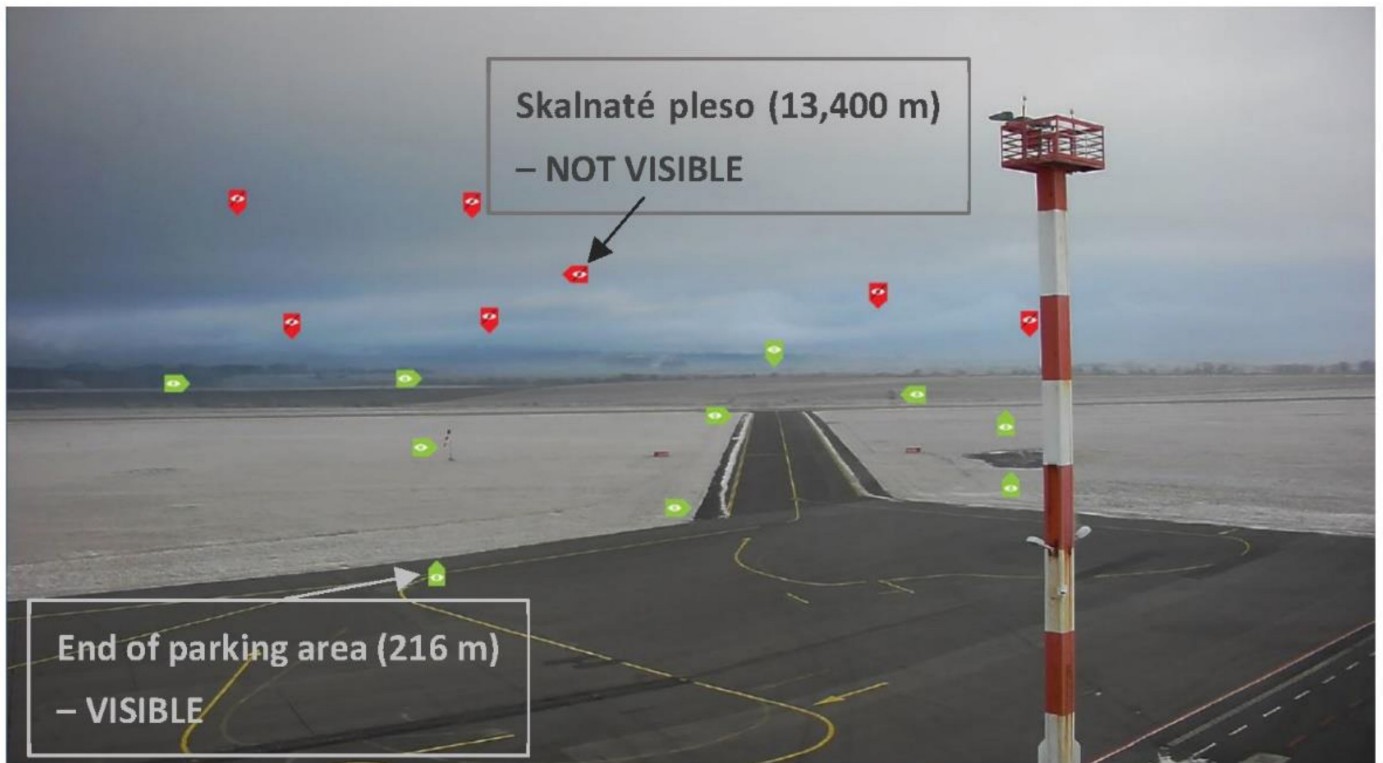

**Figure 6.** Adding markers to estimate the overall prevailing visibility—an example from LZTT, a view in the northern direction.

The data sets that appeared in the validation experiment can be shortly characterized as follows:

- The data set termed as 'standard Automatic Weather Observation System' (AWOS) consists of the data from common sensors for both of the examined meteorological variables. On one hand, it is the visibility sensor Vaisala FS11P. Measurements from this device are available four times a minute. From this data set, 1-min and 10-min averages were calculated. On the other hand, the automatic observations of the cloud heights are supplied by the ceilometer from Vaisala, from which data are available with a 30 s frequency.

- METAR messages are generated at LZTT every half an hour. The only problem of this data set is the missing information on the exact time of the local observation reported in the METAR messages. Based on the oral communication with the observers, this occurs normally 10 min before issuing METAR, i.e., approximately at HH:20 and HH:50. This data set is referred to as 'Local Observations' (LO) corresponding to the fact that it is created by the observers in charge that are present locally at the airport.

- The evaluation of the situations at LZTT that was carried out with the help of the staff of observers at Bratislava will be termed as 'Remote Observations' (RO). The proposed software was installed at the workplace of meteorological observers at LZIB. Observers received training how to validate the LO data set. Then, during the validation period, in addition to their standard work for the Bratislava Airport they also coded METAR messages remotely, for LZTT, using the HMI of the proposed software.

The performance of the proposed system is evaluated by a comparison of the RO, the LO, and the AWOS data observed at the same time, mostly in terms of scatter plots, and also illustrated by a number of use cases.

Note that in the case of visibility observations, an additional experiment was carried out that specifically examined low visibility situations in the period of the cold half-year from September 2018 to April 2019. This additional experiment, unlike the basic evaluation, was carried out in a post-facto way, i.e., a bit longer after the weather happened. Section 3.5 below explains the reasons for this experiment.

## 3. Results

This section presents the results of the validation experiment. It was carried out to compare the local vs. remote observations of the visibility and cloud properties (cloud base height and cloud coverage) in order to assess the reliability of the remote observations by means of the proposed system, and consequently, to answer the research question of whether the local observer could be replaced by the remote one.

### 3.1. The Lowest Cloud Base

The data sets of the lowest cloud base are presented in the form of scatter plots in Figures 7 and 8. In order to allow for a better visual comparison, the y-axes of Figure 7 was cut at 8000 ft (1 ft~0.3 m), instead of displaying the outliers of the AWOS values exceeding 10, 15, and sometimes 24 thousands of feet (associated with LO data from the interval 800 to 3000 ft). Note that cloud base height is expressed in non-metric units (ft), due to the convention of their usage in aviation.

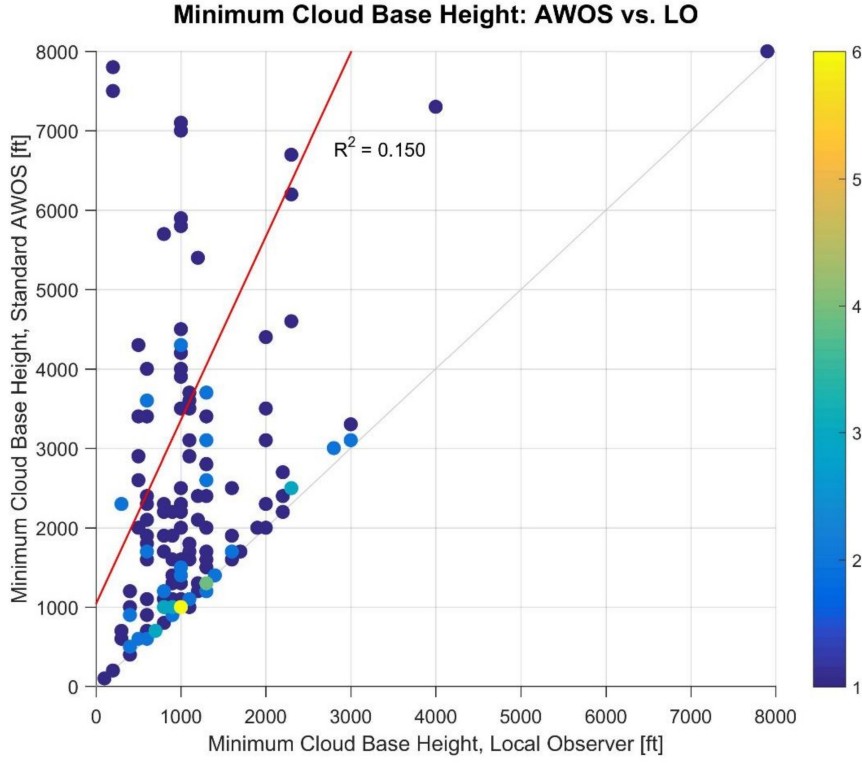

**Figure 7.** Scatter plot of the values of the lowest cloud base determined by the local observers (LO) and the standard automated sensors (AWOS) at LZTT. The color scale indicates the number of occurrence of the given data point.

Figure 8 noticeably indicates a significant improvement in the coefficient of determination of the data sets ($R^2$ = 0.759) when compared to Figure 7 ($R^2$ = 0.150). It is striking that using the RO approach, the outliers clearly diminished, and the data points are scattered relatively tightly around the hypothetical 1:1 straight line.

In other words, Figure 7 points out to the fact that the deviations of the AWOS data set from the ground truth are predominantly positive. This means that the standard AWOS falsely overestimates the actual cloud base heights, and therefore, tend to mislead the pilots with conveying information on better flight conditions than actually are. On the other hand, the RO approach, through the proposed software brings significant improvement in the quality of determining the minimum height of the cloud base, keeping the chance of disinformation of the pilots on minimum.

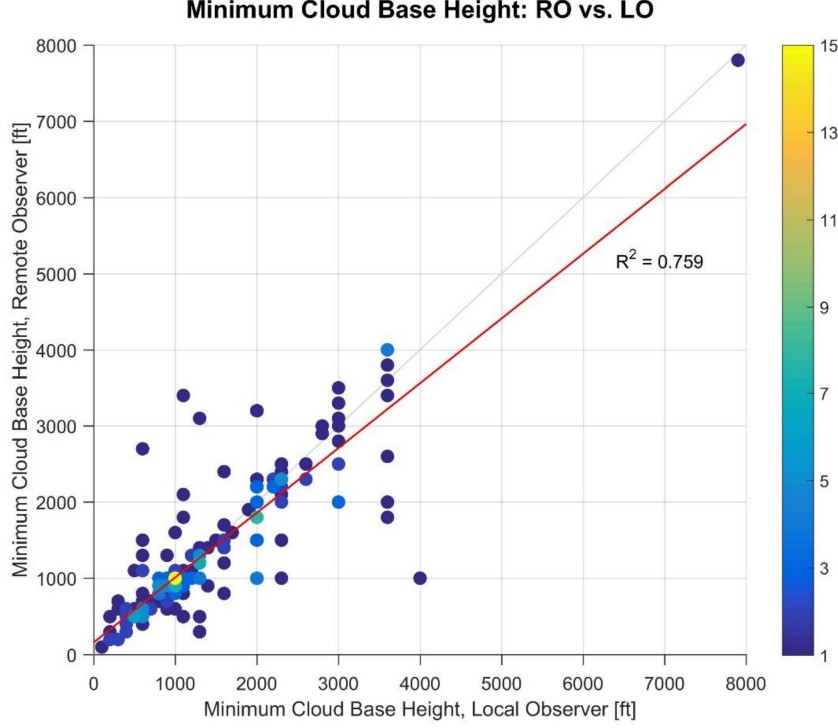

**Figure 8.** The same as Figure 7, but with the lowest cloud base determined by the local observers (LO) and the remote observers (RO).

### 3.2. Maximum Cloud Cover

Beyond the lowest cloud height, the properties of the cloud cover were also evaluated in terms of the maximum cloud cover. The observations carried out by the AWOS, LO and RO approaches were classified on the basis of the standard METAR coding:

- FEW (few)—1 or 2 oktas (eights) of sky are covered by clouds;
- SCT (scattered)—3 or 4 oktas of sky are covered by clouds;
- BKN (broken)—5, 6 or 7 oktas of sky are covered by clouds;
- OVC (overcast)—8 oktas of sky are covered by clouds, i.e., full cloud coverage.

The inter-comparison of the AWOS vs. LO data (RO vs. LO data) is presented in Table 2 (Table 3). Note that the total sum of cases is different in both tables: 181 (227) cases are reported in Table 2 (Table 3). The difference (46) stems in the inability of the standard AWOS to estimate the cloud cover on the basis of a single measurement of the ceilometer, and by means of the associated interpolation methods defined by the manufacturer.

These summaries reveal several characteristic features of the observation methods:

- The AWOS method generally overestimates the maximum cloud cover, by indicating non-empty bins with higher numbers above the diagonal of the observation matrix.
- The AWOS data often differ from the LO data by two or three categories, for instance indicating overcast, whereas the true circumstances are few or scattered. This kind of deviation is rare in the case of the RO data; it only appeared once during the analyzed period of time.

- In line with the previous statements, the percentage of the exact matches (i.e., the sum of figures along the main diagonal of the observation matrix) is much higher in the case of the RO data set. The RO method indicates perfect matches in 80% of all the cases, which is a considerable improvement in comparison with the AWOS observations, with perfect matches only reaching up to 31%.
- If we do not insist on perfect matches, but we allow for a mismatch by one category, the dominance of the RO methodology is still clear: the RO mimics the LO method in 99.55% of all cases, whereas the same statistic is 83% for the standard AWOS.

The validation experiment indicated that the RO method noticeably outperformed the measurements of the standard AWOS, when evaluating both of the perfect or imperfect matches (i.e., a mismatch of one category allowed) in the cloud coverage categories.

**Table 2.** A comparison of the maximum cloud cover values determined by the AWOS and the local observers (LO). The exact matches along the diagonal are indicated in bold.

|  |  | AWOS | | | |
| --- | --- | --- | --- | --- | --- |
|  |  | **FEW** | **SCT** | **BKN** | **OVC** |
|  | FEW | **3** | 5 | 10 | 8 |
|  | SCT | 3 | **2** | 15 | 13 |
| LO | BKN | - | - | **10** | 67 |
|  | OVC | - | - | 4 | **41** |

**Table 3.** The same as in Table 2, but for the remote observers (RO) and the local observers (LO).

|  |  | RO | | | |
| --- | --- | --- | --- | --- | --- |
|  |  | **FEW** | **SCT** | **BKN** | **OVC** |
|  | FEW | **56** | 10 | - | - |
|  | SCT | 2 | **21** | 10 | 1 |
| LO | BKN | - | 2 | **68** | 7 |
|  | OVC | - | - | 13 | **35** |

*3.3. Cloud Cover—Case Studies*

The advantages of using the proposed software, i.e., the RO approach, are demonstrated through four case studies.

The first example (Figure 9) presents a situation with two cloud layers. The LO identified that the lower layer at the height of 6,000 ft was associated with cloud cover SCT (3–4 oktas, Table 4), whereas the second layer at the height of 12,000 ft covered the full sky, i.e., it was OVC (eight oktas). On the contrary, the standard AWOS detected only the second layer of clouds, as the lower layer (visible on Figure 9, as the dark area to the N and NW) was not directly above the ceilometer (located in the SE parts of the airport); thus, the lower layer was invisible to the standard AWOS. Overall, the RO was identical with the LO in terms of the cloud coverage, and they were also close in terms of the cloud base height (Table 4).

**Table 4.** Determination of the maximum cloud cover and the lowest cloud base height using three different approaches at LZTT, on 11 September 2018, at 09:20 UTC. AWOS—Automated Weather Observation System, RO—Remote observer, LO—Local observer, OVC—overcast, and SCT—scattered.

|  | **AWOS** | **RO** | **LO** |
| --- | --- | --- | --- |
| Maximum cloud cover | OVC | SCT | SCT |
| Minimum cloud base [ft] | 12,000 | 7000 | 6000 |

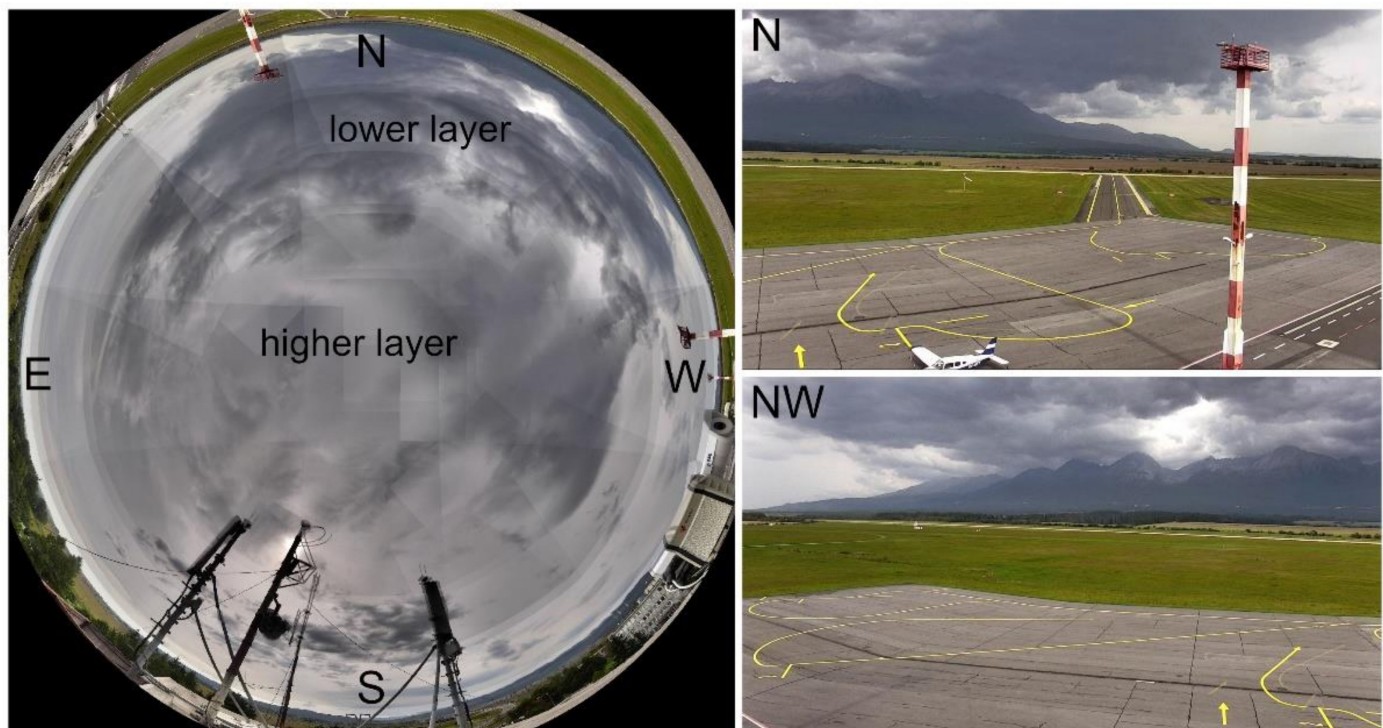

**Figure 9.** Determination of the maximum cloud cover and the minimum cloud base height using cameras at LZTT, on 11 September 2018, at 09:20 UTC.

The second example is associated with the presence of towering cumulus (TCU) up to 16 km from the airport. This type of cloud, together with cumulonimbus (CB), are significant cloud types in aviation; thus, their detection is of a general interest. Both the local and the remote observers identified TCU correctly (Figure 10), with no difference in the height of the minimum cloud base (Table 5). Conversely, the standard AWOS does not have an ability to recognize these types of clouds significant to the airport operation [1]. Nor did the standard AWOS detect the height of these clouds, because they are not in the sky above the ceilometer, but rather on the N-NW horizon. Arrows in Figure 10 also indicate further cloud types (altocumulus and cirrus); however, these are not significant in aeronautics, and they are not indicated in the METAR/SPECI messages.

**Table 5.** The same as Table 4, but for 14 September 2018, at 09:50 UTC. TCU—towering cumulus and CB—cumulonimbus.

|  | AWOS | RO | LO |
|---|---|---|---|
| Maximum cloud cover | - [1] | FEW | FEW |
| Minimum cloud base [ft] | - | 4000 | 4000 |
| TCU or CB | - | TCU | TCU |

[1] No significant clouds.

The third example demonstrates a special case, when, despite the sky being covered significantly, there is a 'hole' in the cloud cover (blue sky) right above the ceilometer. The all sky image in Figure 11 depicts cloud cover BKN (5–7 oktas), which was recognized by both the LO and the RO (Table 6). However, the standard AWOS failed to determine the cloud cover accurately due to a 'hole' in the cloud cover above the ceilometer, and indicated significantly underestimated cloud cover.

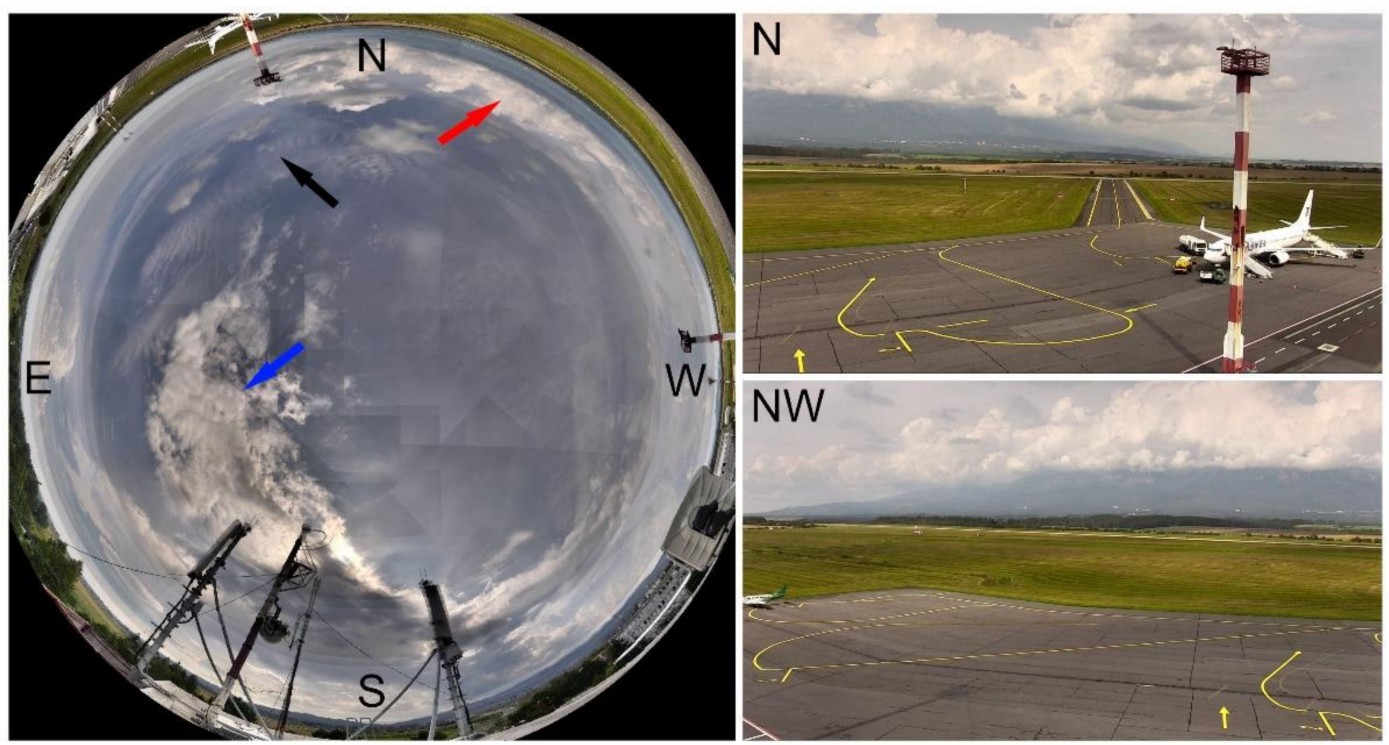

**Figure 10.** The same as Figure 9, but for 14 September 2018, at 09:50 UTC. Arrows denote different cloud types: red—towering cumulus, blue—altocumulus, and black—cirrus.

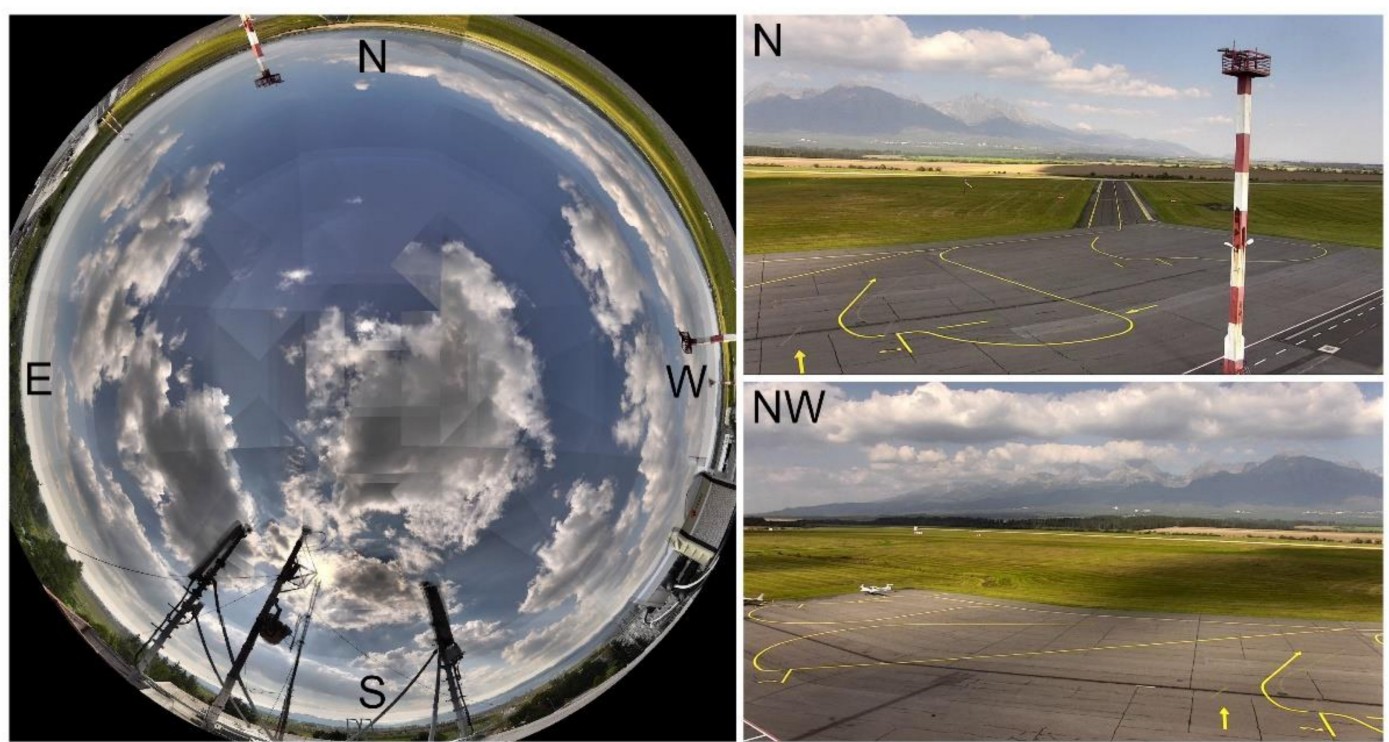

**Figure 11.** The same as Figure 9, but for 21 September 2018, at 09:20 UTC.

**Table 6.** The same as Table 4, but for 21 September 2018, at 09:20 UTC. BKN—Broken.

|  | **AWOS** | **RO** | **LO** |
|---|---|---|---|
| Maximum cloud cover | FEW | BKN | BKN |
| Minimum cloud base [ft] | 6300 | 6300 | 6300 |

The last example practically demonstrates an 'inverse' situation compared to the previous one. Here, the all sky image in Figure 12 clearly shows cloud cover SCT (3–4 oktas), which was recognized both by the RO and the LO (Table 7). However, the standard AWOS failed to determine the accurate cloud coverage, as the clouds appeared statically above the ceilometer position in the SE part of the airport and the device 'did not see' the major clear sky portion.

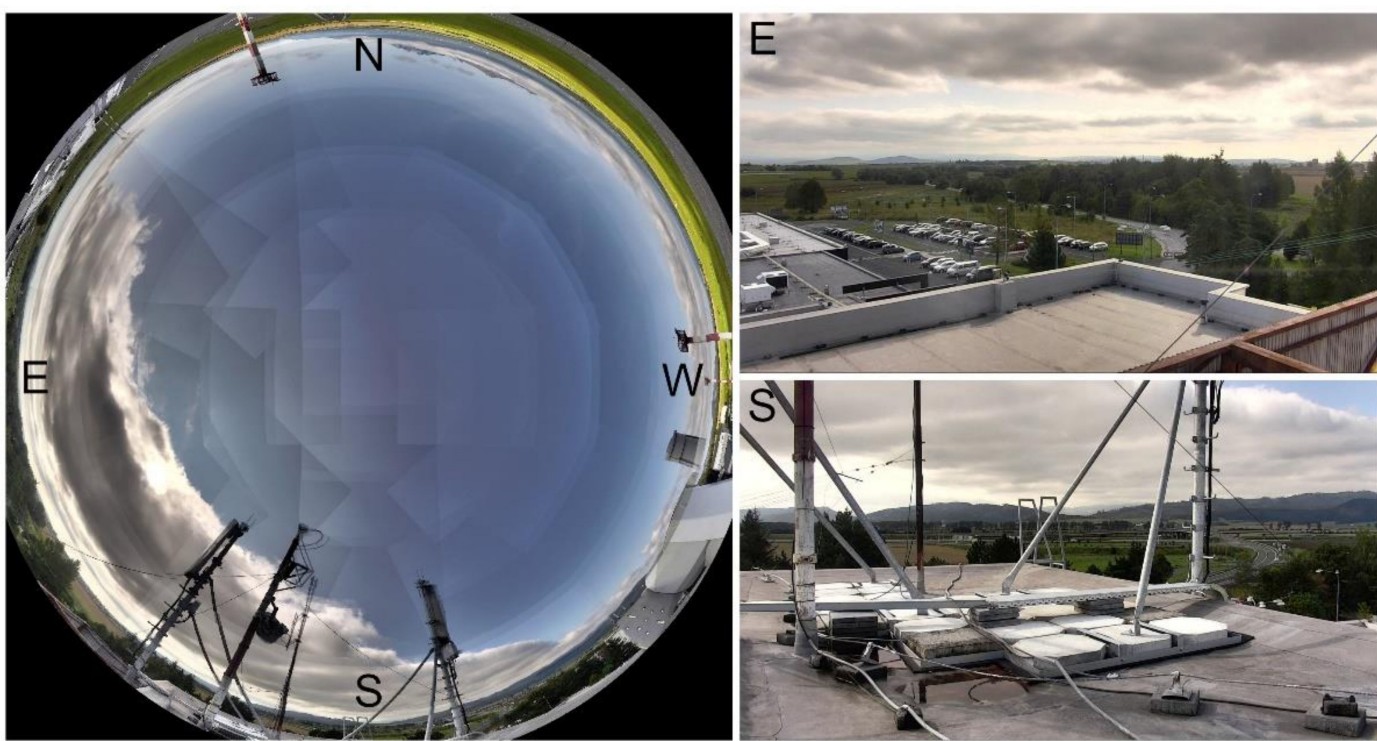

**Figure 12.** The same as Figure 9, but for 27 August 2018, at 07:20 UTC.

**Table 7.** The same as Table 4, but for 27 August 2018, at 07:20 UTC. BKN—Broken, SCT—Scattered.

|  | **AWOS** | **RO** | **LO** |
|---|---|---|---|
| Maximum cloud cover | BKN | SCT | SCT |
| Minimum cloud base [ft] | 1600 | 1600 | 2100 |

### 3.4. Prevailing Visibility—Basic Validation

In the basic validation experiment, the daytime visibility for all the METAR schedule times were parallel assessed (i) by the local observers (LO) at LZTT, and (ii) by the meteorologists of the Slovak Hydrometeorological Institute remotely, by means of the proposed system (i.e., RO). Both data sets are comprised of 245 individual observations.

First, the performance of the standard automated sensors (AWOS) and the novel RO approach were inter-compared, in the light of the LO data set, which is considered as the ground truth. The results are presented in the form of scatter plots (Figures 13 and 14). Note that due to the natural dominance of situations with good visibility, i.e., multiple

occurrence of pairs of [9999; 9999] (211 times in both figures, indicated by yellow color), only a reduced set of pairs of observations are discernible in the scatter plots.

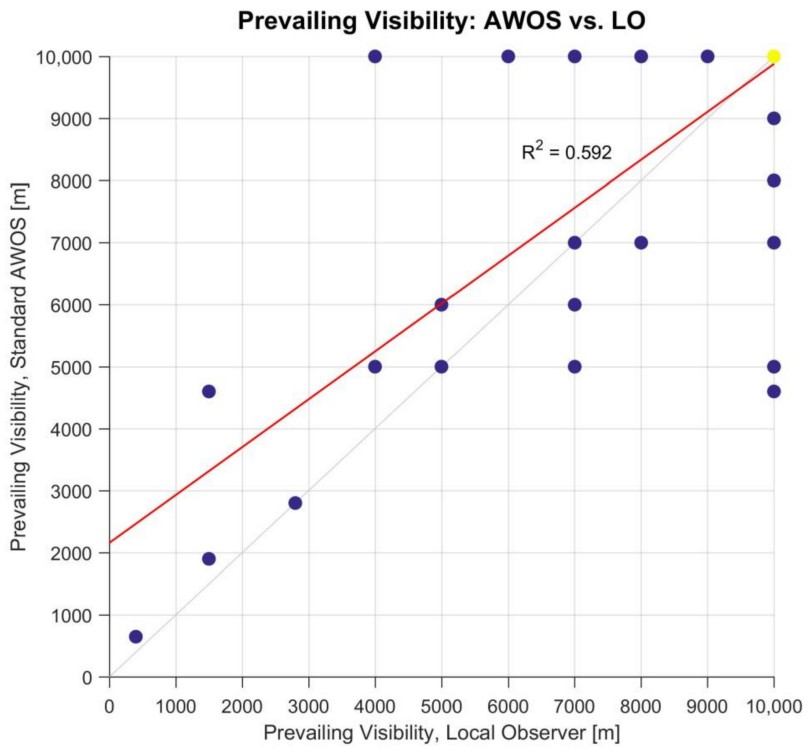

**Figure 13.** Scatter plot of the values of the prevailing visibility determined by the local observers (LO) and the standard automated sensors (AWOS) at LZTT.

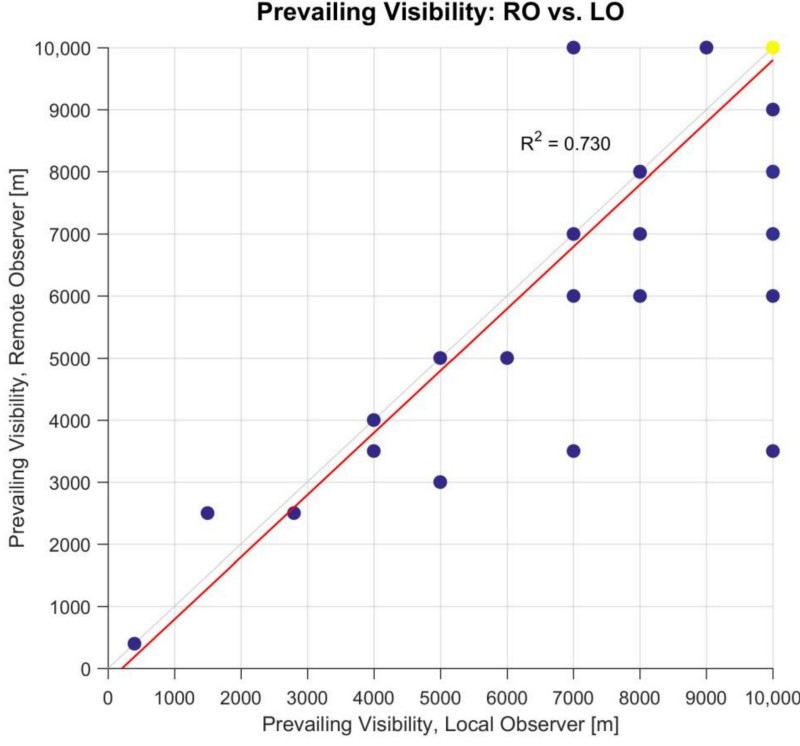

**Figure 14.** The same as Figure 13, but with the prevailing visibility determined by the local observers (LO) and the remote observers (RO).

The scatter plots in Figures 13 and 14 indicate an improvement in the observation accuracy. The comparison of the prevailing visibility is in favor to the RO data set with a correlation coefficient of 0.73, whereas the reference AWOS scenario reached only 0.59.

Secondly, the LO and RO data sets were compared and evaluated directly, according to the criteria of the ICAO [1] 'Operationally desirable accuracy of measurement or observation'. They require the following accuracy for the prevailing visibility:

- $\pm 50$ m up to 600 m;
- $\pm 10\%$ in the range of 600 to 1500 m; and
- $\pm 20\%$ above 1500 m.

Note that to obtain a finer stratification, we decided to split the last visibility interval into two parts: 1500 m to 5000 m, and 5000 m and more, respectively. In the light of these categories, the results of the basic validation experiment are stratified as summarized in Table 8.

**Table 8.** Results of the basic validation experiment for assessing the accuracy of the visibility by the proposed system at LZTT.

| Visibility Range [m] | Number of Situations |
|:---:|:---:|
| <600 | 1 |
| 600–1500 | 2 |
| 1500–5000 | 5 |
| >5000 | 237 |

The most important finding of the summary in Table 8 is the fact that the categories with medium and low visibility ranges are considerably underrepresented—the prevailing visibility dropped below 5000 m only in eight cases (3.3%). Consequently, no generally valid conclusions can be drawn on the basis of such spare data. This is particularly a pity, as the situations with medium and low visibility ranges are the most crucial for the air traffic navigators.

The basic validation experiment was conducted on data set with natural distribution of low visibility situations, and there lies its relevance. However, it also indicated the necessity of carrying out a further, more extensive one, with the focus set mostly on the weather events accompanied by low to medium visibility ranges.

*3.5. Prevailing Visibility—Extended Validation*

The extended validation experiment was targeted on the period of the cold half year from September 2018 to April 2019. This time span was selected because in Slovakia (or, in a broader sense, in Central Europe), it is the period of the year with higher probability of the occurrence of fogs and lowered visibility situations. It was a retrospective (post-facto) analysis, as it was carried out later, during the late spring and early summer of 2019, focusing predominantly on weather events with decreased visibility.

All the situations were assessed by the local professional observers on one hand, and by three independent meteorologists as remote observers on the other. Due to the higher number of remote observers, one can compare the local and remote observations in different ways. The stricter approach (from now onward: 'strict criterion of comparison') is to require meeting the ICAO accuracy conditions by all three remote observers at the same time. A more lenient approach ('loose criterion of comparison') is to declare that it is sufficient that at least one remote observer meets the accuracy conditions. When presenting results, we will prefer the first, stricter approach (if not emphasized otherwise).

The extended validation experiment resulted in figures that are summarized in Table 9. The increase in the number of low visibility situations is clear in comparison to Table 8.

**Table 9.** Results of the extended validation experiment for assessing the accuracy of the visibility by the proposed system at LZTT in the period of September 2018–April 2019.

| Visibility Range [m] | Number of Situations |
|:---:|:---:|
| <600 | 34 |
| 600–1500 | 14 |
| 1500–5000 | 231 |

For each of the 34 weather events from the category with the lowest visibility ranges (i.e., 600 m or lower), i.e., the most adverse ones for the air navigation, a separate detailed analysis was carried out. It turned out that these situations could be classified into five, well-defined categories (Table 10).

**Table 10.** Classification of the situations with the lowest visibility ranges (i.e., 600 m or lower) from the extended validation experiment.

| Category # | Category Description | Number of Situations |
|:---:|:---:|:---:|
| 1 | In the range of the strict ICAO accuracy condition | 17 |
| 2 | Different time of assessing the visibility | 4 |
| 3 | Reporting the visibility sensor data | 2 |
| 4 | Different visibility in different directions | 3 |
| 5 | Different estimation by the local observer | 8 |
| | (Total) | (34) |

Category #1 of Table 10 represents the cases that all meet the strict ICAO accuracy condition, i.e., none of the three values of prevailing visibility assessed by the remote observers differ from the visibility estimated by the local observer by more than ±50 m.

Category #2 of Table 10 is directly related to the lack of the explicitly defined time to determine visibility conditions. Only the time period is defined when the observation should be made, and it is the period of 10 min just before the METAR validity time. Such a flexibility may lead to differences in the reported values of the local vs. remote observers, particularly in cases of rapid changes in meteorological conditions. On the basis of the data from the visibility sensors, we were able to identify situations such as that. For instance, on 9 October 2018, over the course of 20 min (from 06:10 to 06:30 UTC) the visibility increased by more than 1000 m. In case the local observer would evaluate the prevailing visibility a bit earlier, e.g., at 06:18, the final METAR record would be 200 m. If the observer decided to carry out the observation at 06:20, one would get a value of 300 m; and if at 06:22, the visibility would be 600 m.

Category #3 of Table 10 indicates the number of cases, when the local observer reported the data directly from the visibility sensor, instead of his/her own observation. At the time of determining the visibility, each observer has got the current AWOS data (the latest averages of the visibility from a 1-min and a 10-min window) available. It turned out that there were situations where the observer, for reasons that are not entirely clear to us, decided to send the sensor data to the METAR message, even if these data did not correspond one hundred per cent with the real atmospheric situation A possible explanation of such behavior is that the data from the visibility sensor is the only reference point against which the observer can be retrospectively checked in the current practice.

Category #4 of Table 10 relates to meteorological situations with different visibility conditions in different directions. These represent difficulties for the local observer. ICAO defines the prevailing visibility " ... within at least half of the circle of the horizon ... " [1]. Sometimes it is only a question of a few degrees whether visibility for the half of the circle is or is not achieved. In other words, minor differences in the (subjective) perception may result in significant differences in the overall evaluation of the given situation. In cases like that, it really depends only on the observer which value to decide on.

Category #5 of Table 10 counts eight situations that were impossible to classify into categories #1–4.

As the extended validation indicated, there are different known (Categories #2–4) or unknown (Category #5) reasons that might explain the differences in the evaluation of the prevailing visibility during weather situations with decreased visibility. A common feature of the known or at least supposedly known reasons is the attitude of the local professional observers, which will be analyzed in detail in the Discussion section.

### 3.6. Prevailing Visibility—Case Studies

Considerable differences in the perception of the prevailing visibility between the standard AWOS and human observers appear predominantly in situations when spatially inhomogeneous fog covers the target area. Two of such situations were selected as illustrations.

The first example presents fog in patches at LZTT (Figure 15, Table 11). It occurred on 10 October 2018 when the overall synoptic situation was governed by an anticyclone over the Central Europe. Consequently, the sky was clear, and due to the absence of clouds overnight, radiation fog formed over the area of LZTT. The light wind (up to 1–2 m/s) during the night did not disrupt the formation of the radiation fog.

At the time of measurement (04:50 UTC), the standard AWOS was covered by some fog patches; therefore, it indicated medium range visibility of 4200 m (Table 11). On the contrary, the remote observer, with the help of camera images, was able to see the entire horizon, and consequently, was able identify different visibility conditions in different directions, with minimum visibility towards the east (Table 11). The same was observed by the LO. Fog with patches (BCFG) was observed between 02:00 and 5:30 UTC, then it changed to VCFG, i.e., fog in the vicinity, and it disappeared at about 08:00 UTC.

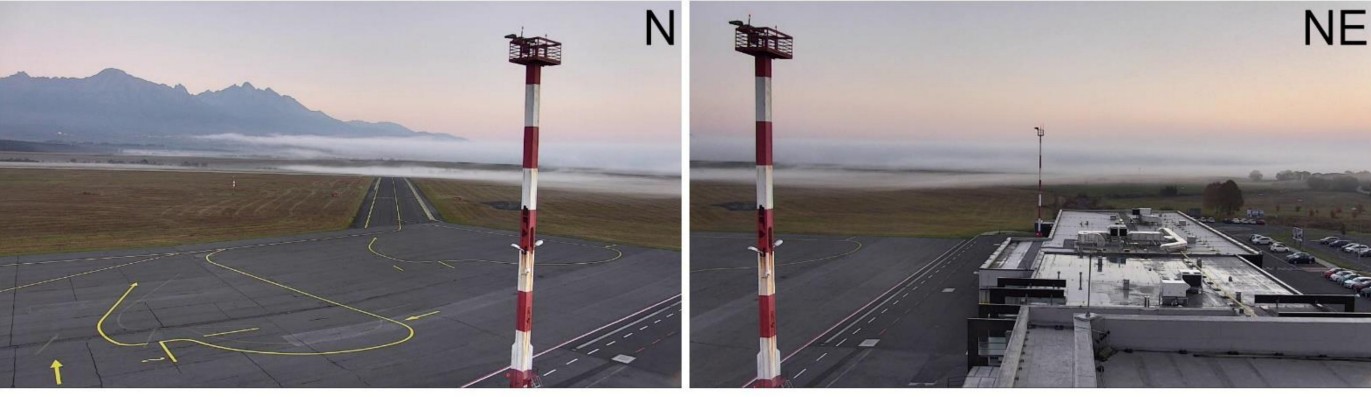

**Figure 15.** Fog in patches at LZTT, on 10 October 2018, at 04:50 UTC.

**Table 11.** Estimation of visibility using three different approaches at LZTT, on 10 October 2018, at 04:50 UTC. BCFG—Fog in patches.

|  | **AWOS** | **RO** | **LO** |
| --- | --- | --- | --- |
| Visibility [m] | 4200 | 9999 1000 to East | 9999 1000 to East |
| Phenomenon | non | BCFG | BCFG |

The second use case demonstrates a meteorological situation with fog of spatially variable density (Figure 16, Table 12). On 28 October 2018, the region of the Central Europe was influenced by a low with a center over the Ligurian Sea, and by an associated warm front, progressing from south to north across Slovakia. The fog at LZTT occurred in this synoptic situation, approximately between 04:00 and 07:00 UTC. Due to the uneven density

of the fog, the standard AWOS estimated the visibility as of 3000 m, and consequently, the fog was categorized as mist. On the other hand, both the local and the remote observers reported worse visibility conditions than the AWOS: both human observers agreed that the prevailing visibility was 700 m with fog as the dominant meteorological phenomenon (Table 12).

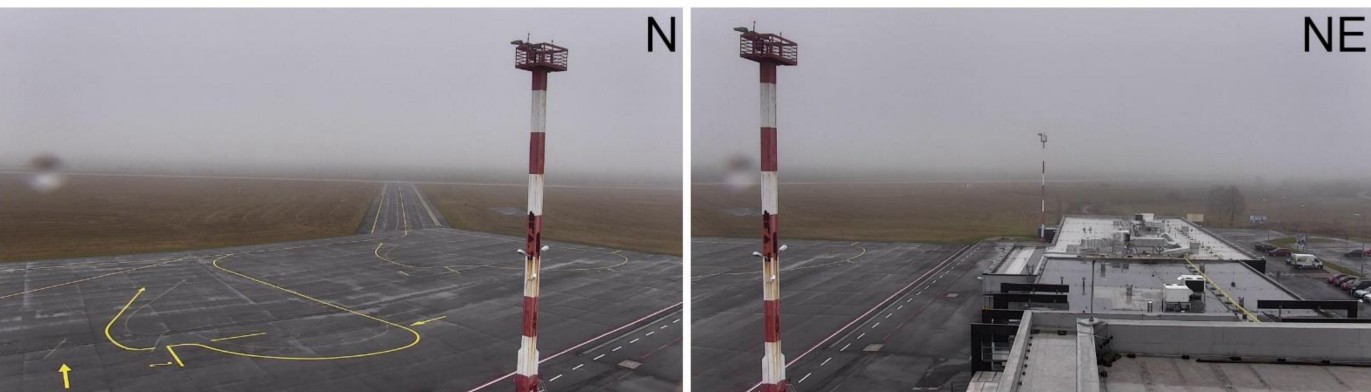

**Figure 16.** The same as Figure 15, but for 28 October 2018, at 06:50 UTC.

**Table 12.** The same as Table 11, but for 28 October 2018, at 06:50 UTC. BR—mist, FG—fog.

|  | **AWOS** | **RO** | **LO** |
| --- | --- | --- | --- |
| Visibility [m] | 3000 | 700 | 700 |
| Phenomenon | BR | FG | FG |

## 4. Discussion

Generally, the validation experiments confirmed our preliminary hypotheses: the remote observer (RO) approach clearly demonstrated better performance than the standard AWOS, and, at the same time, the performance of the RO was comparable with the local observations (LO), both for the estimation of the cloud base height (Figures 7 and 8) and the prevailing visibility (Figures 13 and 14). Nevertheless, the obtained results are far from an ideal relationship between the RO and the LO approaches with $R^2$ close to 1.00, probably due to the following reasons:

(1) The conclusions are drawn from a small sample, and
(2) The whole experiment is based on the hypothesis that the LO represent the ground truth.

Here, the second point is rather questionable. What if we admit that mistakes may also appear on the side of local observers? As the observations of the prevailing visibility and cloud coverage are carried out by human observers, using the tool named 'human eye', and with no other objective measurement devices, the result is inherently biased by subjectivity. However, as far as we know, there is no study that would evaluate this bias in the light of diverse perceptions of individual observers and/or complex conditions for observation and other factors. This is a problem, as ignorance of the error rate of the reference observation reduces the accuracy of the validation of any new system targeted on the determination of these meteorological variables.

The extended validation experiment with the three independent remote observers proved that in spite of the availability of significant technological support (photos with associated reference points that can be checked repeatedly), different professional observers may evaluate a given low visibility situation in a different way, even with a difference of ±50 m in their estimates. Local observers are also human, and they have to cope with the dynamic and unrepeatable character of nature in the absence of the technological support

of the database of camera images. Therefore, we suppose that it is reasonable to question the unmistakable character of the observations carried out by the local professionals.

But how could be this hypothesis verified? Obviously, one could design a long-term experiment with three independent local observers parallel in charge. Nevertheless, it is not a straightforward task to carry out such an experiment, for different serious reasons. Generally, there is the lack of human resources. The minimum requirement is to have five experts to cover the weather service duties in 24/7. To accomplish the task with three local observers operating at the same time, one would need an at least three times larger staff of observers, further extra budget to cover their salaries, and not less importantly, to have sufficient infrastructural background; for instance, to ensure the limitation/blockage of communication of the observers among themselves. Otherwise, it is nearly impossible to evaluate the degree of subjectivity of the professional observers. It is in the future plans of the authors of this article to design and carry out such an experiment.

The extended validation experiment shed light on the fact that the correct character of the observations may also be influenced by the attitude of the local observers, e.g., when they simply copy the data supplied by the standard AWOS or when a minor detail in the process of the evaluation of a weather event results in more significant differences in the overall assessment of the given situation. Furthermore, the loose definition of the time window to carry out the observations may lead to an incorrect evaluation of the current weather situation—as illustrated by the four cases with rapidly changing meteorological conditions, underpinned by the supplementary data from the visibility sensor. Using the proposed system, this type of inconsistency can be easily avoided by regular observations scheduled at a given time point within a period of 10 min before the METAR validity time. One of the further undisputable benefits of the proposed system is its functionality to hold a backup of all the observations from the past. It allows for verifying and eventually modifying the LO at any time, both for the visibility and the cloud coverage.

The small size of the sample (mentioned above in the first bullet point) implies that the findings of the current study should not be overly generalized. They only indicate that the direction of our research is encouraging, and to derive generally valid conclusions, a much deeper and thorough analysis of the performance of the RO approach is required. A new, more elaborated analysis in the near future is expected to cover a period of length of about two years and will include a visual analysis of the camera records of several thousands of weather situations by remote observers.

There is another specific feature of the current analysis, and it is related to the fact that the target destination is surrounded by mountains (the High Tatras in the north). Consequently, Steps #1–3 to determine the cloud base height (described in Section 2.2) can be performed with no limitations. In sites with no mountains or tall country features nearby, however, Step #2 has to be skipped. In such case, the LO is restricted to only follow Steps #1 and #3, and naturally, the same also holds true for the RO. This might lead to different results. Unfortunately, we have not received data from a different airport without mountains or tall features to verify the consequences of (the lack of) the particular settings. Nonetheless, we expect that both the local and the remote observers would be influenced in a similar way, as in the case of a non-mountainous airport, both LO and RO would suffer in the absence of Step #2.

Note that both the LO and the RO should consider the fact that a cloud layer can have different height just above the airport and when it 'touches' the nearby mountains. However, the cloud observations in METAR should be representative of the area within a radius of approximately 16 km of the aerodrome reference point [15]; therefore, possible height differences over these 16 km are not taken into account.

In principle, a very specific case would be an estimation of the cloud base height even without Step #3, i.e., with no ceilometer data. In fact, though, an airport without a ceilometer is a rarity. At less important, non-airport meteorological stations, it is more common to have a human observer without a support of a ceilometer. They even have different, far less precise reporting rules that can be better fulfilled by Step #1, i.e., by

recognizing the cloud types and estimating their typical heights. This is, however, not our case. We target airports, and they tend to be the best-equipped meteorological stations.

The near-future ambition of the team of authors is to upgrade all the subparts of the proposed system of remote observations to a fully automated one. Attempts for automated recognition of visibility were already begun in the 1980s [16,17]. One of the key issues of the automated estimation of visibility is to recognize the contrast of a given object with respect to its background and evaluate it against pre-defined thresholds. To carry out these analyses, data-driven methods, such as deep learning, e.g., convolutional neural networks [4,18], are generally used.

It is expected that the artificial intelligence methods would also help in automated estimation of the cloud coverage. Photographs of the upper hemisphere, e.g., from fisheye-camera-based systems, are mostly utilized in astronomy (for instance, in optimization of closing/opening a telescope depending on the cloudiness, [10–12]) and the energy industry (particularly at solar plantations [13,19]). In our case, the stitched images of the full hemisphere will be used to estimate the sky index [20] and the greyness rate index [21], which again, will be further used to segment the image, based on the Otsu's method of automatic thresholding [22], i.e., to decide which pixels belong to the foreground (clouds) and which ones represent the background (clear sky).

Methods for automated recognition of cloud types from camera images require more sophisticated statistical methods compared to the detection of cloud coverage; these are at research level, however, they generally lack real implementation in practice [23–25].

## 5. Conclusions

The current paper discusses some of the weaknesses of the aeronautical meteorological observations that are related to the necessity of the presence and the experience of the professional meteorological observers directly at the target airports. The difficulties are associated with the meteorological variables, such as visibility, cloud base height, and cloud coverage. Although specific automated sensors (ceilometers, transmissiometers, and forward scatter sensors) exist for the observation of these elements, due to their predominantly point character of measurements they cannot fully emulate a comprehensive way of perception of human observers. The proposed solution to overcome the weaknesses of the observation of these variables is to use a system that, with the help of camera-based images and a user-friendly human–machine interface, allows for remote observation of visibility and cloud properties.

A validation experiment was designed and carried out at the Poprad-Tatry Airport to assess the first performance measures of the remote observations (RO) in comparison with the local observations (LO, declared as ground truth) and the measurements obtained by the standard automatic weather observation system (AWOS). The results of the validation experiment were promising, and indicated that the RO approach was viable. For all the variables (prevailing visibility, maximum cloud coverage, and minimum cloud base height), the RO considerably outperformed the AWOS, and at the same time, the performance of the RO was comparable with the LO approach. The case studies demonstrated that the automated sensors do suffer of shortcomings in certain, predominantly spatially inhomogeneous meteorological situations. These are, for instance, occurrence of several layers of clouds or unevenly covered sky in the case of ceilometers, and fog in patches or fog with spatially variable density in the case of transmissiometers/forward scatter sensors. All the limitations of the automated sensors stem in their point character of measurement. Therefore, they are particularly useful in homogeneous meteorological conditions, and/or for decisions that only require point character of information—e.g., for an estimation of the runway visual range for the certain third of the runway. For a correct estimation of the prevailing visibility, cloud cover, and cloud types, these sensors alone are inadequate; however, the camera-based remote approach to observations seems to be a promising supplement to eliminate the sensors' deficiencies, in terms of the:

- Quality—The proposed solution makes use of high quality camera records for all the eight cardinal directions to determine visibility and the possible presence of dangerous cloud types; and high quality 'stitched' image of the full hemisphere to determine the cloud coverage. All of these materials ensure that estimation of the given meteorological variable is not a point-based one.
- Objectivity—The remote method of observation offers a number of annotated markers to determine visibility (considerably more markers than the local observers have), and a mesh grid of concentric circles to determine cloud cover in an objective way. The degree of objectivity is further increased by the fact that the records of the weather situations are instantaneously archived in a database, and thus, they can be repeatedly double-checked at any time.
- Efficiency—With the proposed remote observing systems installed, the physical presence of an observer at the site of interest is not necessary. The observer can estimate the meteorological variables from a centralized office effectively, even for a number of target sites. This option is particularly useful in COVID-19 and post-COVID-19 conditions.

It is not meant that the proposed remote observation methods should entirely replace the existing sensors. Instead, a combination of the standard and the novel methods would result in synergies that would bring benefits for all the parties involved in providing higher quality aeronautical services.

Our experience to date confirms the knowledge that remote access will allow more efficient use of available human resources, improve employee workload management, and improve work organization up to the so-called home office for selected professions that are and remain a challenge in the current pandemic era, but also in the post-COVID-19 period. We can also state that the proposed solution has the potential to improve meteorologists' decision-making processes and reduces the scope for human error and subjective data evaluation. Finally, it may enable more efficient use of funds for the provision of this service for professional staff, but also for the necessary infrastructure. We have set aside a cost-benefit analysis of our proposed innovative approach and technology for a companion paper.

**Author Contributions:** Conceptualization, J.B.; methodology, J.B.; software, L.I.; validation, I.B. and L.I.; formal analysis, L.I. and I.B.; investigation, L.G.; resources, J.B.; data curation, L.I.; writing—original draft preparation, L.G.; writing—review and editing, L.G., J.B., L.I. and M.K.; visualization, L.I.; supervision, L.G. and M.K.; project administration, J.B.; funding acquisition, J.B. and M.K. All authors have read and agreed to the published version of the manuscript.

**Funding:** This research was funded by the research project APVV-20-0571 'Intelligent Cloud Workflow Management for Dynamic Metric-Optimized Application Deployment (ICONTROL)' of the Slovak Research and Development Agency, and the SESAR Joint Undertaking under the European Union's Horizon 2020 Research and Innovation Programme under the grant agreement No. 733121 (Solution PJ04-02: 'Enhanced Collaborative Airport Performance Management').

**Institutional Review Board Statement:** Not applicable.

**Informed Consent Statement:** Not applicable.

**Data Availability Statement:** Not applicable.

**Acknowledgments:** We would like to thank to the staff of local observers at the Bratislava Airport (11 people) who acted as remote observers in evaluation of the weather conditions remotely at the Poprad Airport.

**Conflicts of Interest:** The authors declare no conflict of interest.

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
