# Peer review of "A Novel Camera-Based Approach to Increase the Quality, Objectivity and Efficiency of Aeronautical Meteorological Observations"

_applsci, doi:10.3390/app12062925_

Round 1

Reviewer 1 Report

This paper is very interesting for a reader and proposes camera based technique for observations. The option to get the remote observations is a big highlight in our tough times.

Author Response

Response to Reviewer #1

We would like to thank to Reviewer #1 for having read our manuscript, and for his encouraging words. Since there were no questions from him, we cannot write any longer response…

Reviewer 2 Report

This manuscript describes a semi-automatic meteorological observation system aimed for aeronautics, where a remote observer evaluates and reports using data from camera images. My main concern is the lack of details regarding how those measurements are currently taken, and how they are estimated by the human observers. Specifically, what is the current standard way of reporting cloud coverage, visibility, cloud base height?  The following information is not clear about the tests:

  • How do RO and LO estimate cloud base heights?
  • How does AWOS estimate cloud cover?
  • How does LO estimate cloud cover?

Additionally, I would like to know how common "local observers" are. Below are my specific comments regarding typos and parts that need further explanation:

  1. The phrase "on the other hand" is used incorrectly at multiple places. 
  2. A very brief description of the SESAR project is needed. Maybe 1-2 sentences explaining the goal and how it relates to this paper, i.e. if this is a continuation.
  3. Line 27: "in the terms of " -> "in terms of the "
  4. Line 65-67: "The proposed solution has the potential to improve the decision-making processes of meteorologists and to reduce the scope for human error and subjective data evaluation " It is not clear how manual analysis remotely by a person improves subjective data evaluation compared to in-place analysis by a person.
  5. Line 75: "bene-fit" --> "benefit"
  6. Line 113: "On the other hand" --> "However"
  7. Line 130: "local measurement of air between the transmitting and receiving heads", does "local" refer to "short distance" here? If yes, is it possible to define the range? If not, it is not clear, please clarify what that refers to.
  8. Line 209: "to remote observer" --> "to the remote observer"
  9. Table 1: Whay are there different number of markers for LO and RO?
  10. Line 318: "data_set" --> "data set"  
  11. Line 465-466: "schedule times was parallel assessed" -->"schedule times were assessed in parallel"
  12. Lines 488-491: How is it assessed whether the accuracy conditions are met or not?
  13. Line 653: "also" used twice, unnecessarily.
  14. Line 683: "in-stead" --> "instead"

In summary, I believe the authors need to make it clear what the standard way of measuring/reporting currently is and in what parts their system is different other than taking pictures.  

Author Response

Response to the comments of Reviewer #2

We would like to thank to Reviewer #2 for his useful comments that helped strengthen the manuscript.

We are going to reply to the suggestions/question of Reviewer #2 in line. All the Line numbers refer to the submitted (original) manuscript and not the revised one.

This manuscript describes a semi-automatic meteorological observation system aimed for aeronautics, where a remote observer evaluates and reports using data from camera images. My main concern is the lack of details regarding how those measurements are currently taken, and how they are estimated by the human observers. Specifically, what is the current standard way of reporting cloud coverage, visibility, cloud base height? The following information is not clear about the tests:

  • How do RO and LO estimate cloud base heights?

We decided to add (to the beginning of Section 2.2) the following description related to the standard procedures of the observation of cloud properties (also in line with the major comments of the third Reviewer):

‘Before describing the novelties of our proposed system, one has to be familiar with the standard procedures of estimating the cloud properties by the meteorological observers. Cloud base height is currently observed at airports WORLDWIDE in 3 steps:

  • The human observer looks to the sky, and recognizes different cloud layers and cloud types, which indicate initial information of their typical height.
  • If there are some tall country features nearby, the observer utilizes them in a more precise estimation of the cloud base height. If such tall features are not available, this step is skipped.
  • Inside his workroom, the observer checks the ceilometer measurements of the cloud base height, and although they are only point measurements, he utilizes them in a combination with steps 1) and 2) to get as comprehensive information on the cloud base height as possible.

In our system, the remote human observer can perform steps 1) and 2) remotely by cameras and step 3) by a remote access to the ceilometer data. ‘

  • How does AWOS estimate cloud cover?

A ceilometer is able to directly measure only in a single point, i.e., above itself. The cloud cover is then estimated by algorithms.

The algorithms used worldwide in AWOSes for an automatic calculation (or better to say, estimation) of the cloud cover differ in methods and/or complexity; however, in principle their core is as follows: The algorithm takes data from the ceilometer over some time period, usually 20-30 minutes. Then it makes two assumptions:

  • clouds are homogenously distributed over the sky;
  • the clouds move over ceilometer.

Then, if in the past 30 minutes there is a 10 minute interval with clouds above ceilometer and a 20 minute interval with no cloud detected above the ceilometer, the algorithm infers cloud cover as 1/3 of the sky.

  • How does LO estimate cloud cover?

The observer looks at the sky, and estimates the cloud cover by comparing parts covered by clouds and parts with clear sky. There are some additional instructions, like ‘imagine if all the clouds moved together without holes, and compare this with the remaining sky’.

Additionally, I would like to know how common "local observers" are.

It is very common. The vast majority of the countries worldwide, that report airport weather internationally, have local human observer.

Below are my specific comments regarding typos and parts that need further explanation:

  1. The phrase "on the other hand" is used incorrectly at multiple places.

The phrase ‘on the other hand’ appeared 7 times in the manuscript. We tried to replace 4 of them with some proper synonyms.

  1. A very brief description of the SESAR project is needed. Maybe 1-2 sentences explaining the goal and how it relates to this paper, i.e. if this is a continuation.

We modified the original sentence at Lines 174-764

‘The basics of the proposed system were laid down in the framework of a SESAR program, namely in the project ‘Pj.05-05 Advanced Automated MET System’.’

as follows

‘The basics of the proposed system were laid down in the framework of a SESAR program. The SESAR project Pj.05 was dedicated to Remote Towers, i.e., to developing air traffic services by a remote Air Traffic Controller. Following this, the project ‘Pj.05-05 Advanced Automated MET System’ launched our experiments in an accompanying service, termed as Remote (meteorological) Observer.’

  1. Line 27: "in the terms of " -> "in terms of the "

Corrected.

  1. Line 65-67: "The proposed solution has the potential to improve the decision-making processes of meteorologists and to reduce the scope for human error and subjective data evaluation" It is not clear how manual analysis remotely by a person improves subjective data evaluation compared to in-place analysis by a person.

Cloud cover is determined by the local observer with a look at the sky, as described above. Estimation of the cloud cover on the basis of a composite image of the whole sky, displayed on the screen of a monitor (where several concentric circles and delineation of the hemisphere into oktas is added) is, however, more objective. Moreover, images of the hemisphere are stored in a database, and this allows for retrospective verification of any weather situation.

In the case of visibility observation, the proposed system offers more markers than the local observer has got on his map. Furthermore, the prevailing visibility is estimated on the basis of an objective software algorithm (which is based on the complete set of the markers with the status ‘visible’ or ‘not visible’), instead of a subjective decision in the mind of the local observer.

  1. Line 75: "bene-fit" --> "benefit"

Corrected.

  1. Line 113: "On the other hand" --> "However"

Corrected.

  1. Line 130: "local measurement of air between the transmitting and receiving heads", does "local" refer to "short distance" here? If yes, is it possible to define the range? If not, it is not clear, please clarify what that refers to.

The reviewer is right, the word ‘local’ refers to short distance. The referred sentence has been corrected accordingly (new insertions are underlined).

Forward scatter sensors are suitable at measuring the transparency of the atmosphere; nevertheless, they only supply a short distance measurement of air between the transmitting and receiving heads (on the order of a few tens of cm, according to the manufacturer).

  1. Line 209: "to remote observer" --> "to the remote observer"

Corrected.

  1. Table 1: Why are there different number of markers for LO and RO?

The difference in the number of markers for LO and RO stems in practical reasons. The number of markers for LO is taken directly from the sketch/map that is used at the airport in operational praxis. Camera system allows RO not to be overwhelmed by additional markers to LO’s ones and still be able to practically use them.

  1. Line 318: "data_set" --> "data set"

Corrected.

  1. Line 465-466: "schedule times was parallel assessed" -->"schedule times were assessed in parallel"

Corrected.

  1. Lines 488-491: How is it assessed whether the accuracy conditions are met or not?

We strictly compared the values derived using cameras with those observed by the LO. In other words, when LO observed 500 m, and RO between 450 m and 550 m, it was within the accuracy conditions (set by rule ±50 m up to 600 m). Otherwise it was out.

  1. Line 653: "also" used twice, unnecessarily.

Corrected.

  1. Line 683: "in-stead" --> "instead"

Corrected.

In summary, I believe the authors need to make it clear what the standard way of measuring/reporting currently is and in what parts their system is different other than taking pictures. 

We reflected the standard way of measuring in our replies above.

The aim of the research was to find out whether the dependence of airport observations on local human presence can be reduced. In the case of a number of meteorological variables like pressure, temperature, wind, humidity this is already a resolved question since the corresponding specific measuring devices can work automatically and precisely. In case of clouds and visibility, however, still the local human is preferred in the majority of cases. Yes, taking pictures is at the core of this effort. But the devil is in the details…, e.g., when, how, with what tools to process, and how to present it to the users, and not less importantly, how to make their interaction with the system smooth, quick and beneficial. And finally, one of the big questions is, how the proposed procedure is comparable to and compatible with the current procedures, which are being used decades.

Reviewer 3 Report

This paper submitted to the Applied Sciences journal (published by MDPI), by Bartok, Ivica, Gaál, Bartoková, and Kelemen, entitled ‘A novel camera-based approach to increase the quality, objectivity and efficiency of aeronautical meteorological observations’ is a well-written paper that should be published after sufficient revision. The problem is an important one for aviation, and an interesting one, from an applied sciences point of view. The use of the English language and grammar is nearly perfect. I have a few general observations about the paper and a number of specific ones.

———
The general observations include:

A) For the problem of estimating the (prevailing) cloud-base height (for clouds that are not of the fog type), this photographic technique seems be limited to airports with mountains or hills or tall buildings nearer than the horizon. This will allow the cloud bases to be lower than those tall features, and the lack of visibility of the image markers will allow the estimation of the lowest cloud height. The authors chose an airport with nearby hills or mountaintops, which allow this study to proceed. The authors need a discussion of how cloud height would be estimated without tall features nearby.

B) Without initially reading the abstract and introduction very carefully, I originally expected that this work would be at least partially about automated computer-vision studies of this problem. But such studies have been deferred to future work and future publications, as discussed in the last paragraphs of the Discussion section. Instead, the focus is on manual human remote-observer studies of the sky images, which indeed is a worthwhile and important study. It should be made abundantly clear in the abstract and the introduction that the automated computer-vision studies of this problem are not the focus of the current paper.

C) I am very interested in the authors getting this published, but I have some small amount of concern about the declaration of ‘no conflicts of interest’ at the end of the paper. I am no expert at all in this issue of conflicts of interest, but perhaps since the authors’ primary affiliation is a private company (MicroStep-MIS), maybe they have applied for some patents or they have some other private financial interest in this project, which might somehow constitute a conflict of interest? Furthermore, would publication of this work interfere with the applicability of any patents? The authors have had public funding for this work from national and European public funding agencies, so maybe these concerns of mine are already addressed in their funding applications.

D) Are the remote observations done in real-time for this study, as the weather is happening? Or are the remote observations done in a post facto way, a few days or longer after the weather happened?
     This should be made clear.

E) If the authors don’t name the local observers, I think the authors should at least name the remote observers who are the experts in remote observations in this study (perhaps in the acknowledgements). Are any of the remote observers or the local observers included within the author list?

F) Wouldn’t it be useful to have a remote observer station at the desk of the local observer? This could help substantially with removing the probable biases of the local observers. They could just check their real visual observations of visibility and cloud-base height, for example, by comparing those observations to what they could estimate from the remote-observer station’s computer display.
      And they could retrospectively go back and look at some of their past local, visual estimates.

G) The stitching of the sub-images is not described technically at all in this work. And there are visible boundaries of the sub-images. Is it desirable or undesirable that the boundaries of the sub-images not be visible to the observer? Is the stitching the same way for each set of sub-images, with constant positions of the edges of the sub-images? If so, what happens if the pointing of the robotic camera is a little bit different than the preset pointing angle? What happens when one sub-image is darker than the neighbor? Which one goes on top of the other?

H) The 61 images are acquired by the robotic camera, and there is some time delay between each image. How long does the cycle of image acquisition take, for the 61 images? Is the length of this acquisition time a significant issue?

I) For the Lambert azimuthal equal-area projection, is this the best projection to use? This choice needs more motivation. It might be the best, but please convince me.

J) Do local human observers really use/count the markers referred to in Table 1 and Figure 6?

———

Specific observations
1) The term ‘forward scatters’ seems like a slang word. Maybe ‘forward-scatter sensors’ would be more formal?

2) Figure 7: ‘The color scale indicates the density of the data points.’ I don’t understand what this density of data points means.

3) Figs 7-8: Some of the circles appear as chopped-off circles, since they have a straight edge. Maybe this needs to be fixed?

4) Is the total number of entries in table 2 and table 3 different? Maybe write down the total, and explain the difference?

5) Fig. 9:  If it’s overcast at 12,000 ft, and if it’s scattered clouds at 6,000 ft, shouldn’t an all-sky monitor like LO or RO say that it’s overcast? Table 4 says that LO and RO say that it’s scattered clouds. Or are we only concerned about low clouds?

6) Table 10: These results for less than 600m are the only results from the extended validation experiment presented here. I realize that these short distances are more critical distances, but there are a lot more observations at mid-distances in Table 9, which might allow better statistics or better understanding.

7) The authors didn’t use a fisheye lens instead of the robotic camera with stitching. This choice can be motivated more.

8) Citations needed for the consensus stated on line 623.

9) Complaining about lack of grant-proposal approval on line 650 is not the best thing to write in a journal article.

10) Line 670: talking about what is underway at the time of manuscript preparation is not really the most appropriate thing to write about in a journal article. Maybe just refer to ‘future work’ or ‘ongoing work’, etc.

11) Line 676: do the authors mean refs. 11 & 12, instead of refs. 10 & 12?

12) Line 694: maybe change ‘attempted to discuss’ to ‘discusses’ or ‘discussed’?

13) Line 753: maybe change ‘for independent research (paper)’ to ‘for a future paper’ or ‘for a companion paper’.

Author Response

Response to the comments of Reviewer #3

We would like to thank to Reviewer #3 for his very detailed review and his time dedicated to a thorough analysis of some weaker points of our manuscript. We really appreciate his efforts since answering his questions and clarifying some imprecise formulations of our analysis certainly raises the quality of our submitted manuscript.

We are going to reply to the suggestions/question of Reviewer #3 in line. All the Line numbers refer to the submitted (original) manuscript and not the revised one.

A) For the problem of estimating the (prevailing) cloud-base height (for clouds that are not of the fog type), this photographic technique seems be limited to airports with mountains or hills or tall buildings nearer than the horizon. This will allow the cloud bases to be lower than those tall features, and the lack of visibility of the image markers will allow the estimation of the lowest cloud height. The authors chose an airport with nearby hills or mountaintops, which allow this study to proceed. The authors need a discussion of how cloud height would be estimated without tall features nearby.

We decided to add (to the beginning of Section 2.2) the following description related to the standard procedures of the observation of cloud properties (also in line with the major comments of the second Reviewer):

‘Before describing the novelties of our proposed system, one has to be familiar with the standard procedures of estimating the cloud properties by the meteorological observers. Cloud base height is currently observed at airports WORLDWIDE in 3 steps:

  1. The human observer looks to the sky, and recognizes different cloud layers and cloud types, which indicate initial information of their typical height.
  2. If there are some tall country features nearby, the observer utilizes them in a more precise estimation of the cloud base height. If such tall features are not available, this step is skipped.
  3. Inside his workroom, the observer checks the ceilometer measurements of the cloud base height, and although they are only point measurements, he utilizes them in a combination with steps 1) and 2) to get as comprehensive information on the cloud base height as possible.

In our system, the remote human observer can perform steps 1) and 2) remotely by cameras and step 3) by a remote access to the ceilometer data. ‘

Note that at an airport with mountains (tall objects…) nearby, both local and remote human observers can follow all three steps. Conversely, at a lowland airport, both local and remote observers can only rely on steps 1) and 3). Therefore, it seems to us that we do not benefit of any specific advantage when selecting a mountainous airport.

The Poprad Airport was selected since we have already had some infrastructure installed there. Currently, however, we are building and developing a new system at another airport (for instance, to ensure more data for the automatic recognition), which is a lowland-type airport.

B) Without initially reading the abstract and introduction very carefully, I originally expected that this work would be at least partially about automated computer-vision studies of this problem. But such studies have been deferred to future work and future publications, as discussed in the last paragraphs of the Discussion section. Instead, the focus is on manual human remote-observer studies of the sky images, which indeed is a worthwhile and important study. It should be made abundantly clear in the abstract and the introduction that the automated computer-vision studies of this problem are not the focus of the current paper.

In line with the Reviewer’s concerns, we modified the abstract at two positions, and a clarified the goals of our research in a paragraph in the Introduction section. The new insertions are underlined.

Abstract, Line 19: ‘We present manned (and not automated) observations from a remote center […]’

Abstract, Line 26: ‘the camera-aided remote human approach to observations […]’

Introduction, Lines 177-180:

‘The aim of the paper is to describe new possibilities of observing the prevailing visibility and cloud coverage / height using a camera-aided observation system, which does not necessarily replace, but effectively and synergistically complements standard observations. Note that the proposed solution is not a fully automated, computer-vision based one. The developed system still relies on the perception, knowledge and experiences of the human observers, though via the assistance of advanced technological utilities.’

C) I am very interested in the authors getting this published, but I have some small amount of concern about the declaration of ‘no conflicts of interest’ at the end of the paper. I am no expert at all in this issue of conflicts of interest, but perhaps since the authors’ primary affiliation is a private company (MicroStep-MIS), maybe they have applied for some patents or they have some other private financial interest in this project, which might somehow constitute a conflict of interest? Furthermore, would publication of this work interfere with the applicability of any patents? The authors have had public funding for this work from national and European public funding agencies, so maybe these concerns of mine are already addressed in their funding applications.

The whole team of co-authors declare, with clear conscience, that there are no conflicts of interest and no patents have been or will be applied in relation with the published topic of the remote observer system. Our general aim is to support the potential future business activities of our company by a scientific paper approved by the research community, i.e., in the form of a publication in a peer-reviewed journal.

D) Are the remote observations done in real-time for this study, as the weather is happening? Or are the remote observations done in a post facto way, a few days or longer after the weather happened? This should be made clear.

In the first paragraph of Section 3.5 (Lines 510-515) where the reason for carrying out the extended validation experiment were explained we indeed mentioned that it had been a retrospective (with the Reviewer’s words ‘post facto’) analysis. On the other hand, we forgot to include a similar statement in relation with the basic experiment. To correct this mistake, we inserted the following sentences (underlined) in Section 2.4 (Data Sets and Methods):

Lines 305-307:

The basic evaluation of the proposed system took place between November 23, 2018, and December 18, 2018, at the Poprad-Tatry Airport (LZTT), with the remote assistance of the professional meteorological observers on duty at Bratislava Airport (LZIB). It was a real-time experiment, i.e., the professional observers at Bratislava carried out parallel observations both at LZIB and LZTT as the weather was happening.

Lines 334-337:

Note that in the case of visibility observations, an additional experiment was carried out that specifically examined low visibility situations in the period of the cold half-year from September 2018 to April 2019. This additional experiment, unlike the basic evaluation, was carried out in a post-facto way, i.e., a bit longer after the weather happened. Section 3.5 below explains the reasons for this experiment.

E) If the authors don’t name the local observers, I think the authors should at least name the remote observers who are the experts in remote observations in this study (perhaps in the acknowledgements). Are any of the remote observers or the local observers included within the author list?

We appreciate this note from the Reviewer. While strictly focusing on the research aspects of the manuscript, we simply forgot to thank for the assistance of those people. The Acknowledgment section, therefore, was modified as follows:

Acknowledgments: We would like to thank to the staff of local observers at the Bratislava Airport (11 people) who acted as remote observers in evaluation of the weather conditions remotely at the Poprad Airport.’

F) Wouldn’t it be useful to have a remote observer station at the desk of the local observer? This could help substantially with removing the probable biases of the local observers. They could just check their real visual observations of visibility and cloud-base height, for example, by comparing those observations to what they could estimate from the remote-observer station’s computer display. And they could retrospectively go back and look at some of their past local, visual estimates.

Thank you, we will consider this idea in our continuing research plans. We did not want to disturb the local observers in any way, so that the experiment would not be influenced. But, with a proper experimental design, this idea could certainly help.

G) The stitching of the sub-images is not described technically at all in this work. And there are visible boundaries of the sub-images. Is it desirable or undesirable that the boundaries of the sub-images not be visible to the observer? Is the stitching the same way for each set of sub-images, with constant positions of the edges of the sub-images? If so, what happens if the pointing of the robotic camera is a little bit different than the preset pointing angle? What happens when one sub-image is darker than the neighbor? Which one goes on top of the other?

We did not want to go deeply in technicalities of image stitching; however, we decided to add the following description to Section 2.2 (after the originally second paragraph at Line 239):

‘The procedure of accurate image stitching needs special attention. Presets for every sub-image are always the same as the camera with the rotator were well fixed on the same place for the whole duration of the research. Countermeasures against possible differences from the pre-defined preset positions have been done. The camera with the rotator have a proclaimed operational wind load durability 216 km/h (135 mph), and during the research operation, no considerable wind gusts have been observed. 1-2 pixel movements could possibly be caused by returning of the camera to the same preset position from different side in the case of power supply outage, but such a difference is negligible for the final output and its usage. Overlapping parts of the sub-images are being averaged if an image is darker than the neighboring one. ‘

Yes, the boundaries between the sub-images are visible on some stitched images. In fact, in our research developments carried out lately, these boundaries have already been removed, but in order to keep the consistency of the research and its results, all graphical outputs for the paper have been created by the ‘old’ method, i.e., using images with boundaries.

H) The 61 images are acquired by the robotic camera, and there is some time delay between each image. How long does the cycle of image acquisition take, for the 61 images? Is the length of this acquisition time a significant issue?

With settings used in the experiment, the entire cycle of acquiring the set of 61 images takes under 3 minutes. In general, there are 2-3 second gaps between two subsequent shots, whereas vertical shifts at the end of each azimuthal rotation cost a few more seconds. Usually it is not an issue, except for the cases when clouds move very quickly. As indicated at the previous point, we are experimenting with a lower amount of pictures, using lenses with a broader field of view, and using these settings, the cycle of shooting takes around 1 minute.

I) For the Lambert azimuthal equal-area projection, is this the best projection to use? This choice needs more It might be the best, but please convince me.

It MUST be equal area, because the human is supposed to estimate cloud coverage fraction, which is essentially comparing the covered area to the whole area.

It does not need to be the Lambert projection. In this case, we simply selected one from several possible projections.

J) Do local human observers really use/count the markers referred to in Table 1 and Figure 6?

Yes, this is a standard way at airports worldwide. The observers have got a sketch or a photo-panorama with annotated visibility points.

1) The term ‘forward scatters’ seems like a slang word. Maybe ‘forward-scatter sensors’ would be more formal?

We agree with the suggestion. The term ‘forward-scatter sensors’ has been replaced 7 times in the manuscript.

2) Figure 7: ‘The color scale indicates the density of the data points.’ I don’t understand what this density of data points means.

Figures 7-8 (and 13-14) were created by a Matlab subroutine, in order have a nice colorful scatter plot that also conveys some information on the multitude of data points that cover each other. Nevertheless, our original solution was not perfect and we agree with the Reviewer that the term ‘data density’ is somehow vague, and would need further clarification.

In the reviewed manuscript, we simplify the graphical representation of the cloud of data points and the corresponding color scale. Instead of data density, the color scale simply indicates the number of occurrence of the individual data points. For instance, in Figure 7, the data point of [1000,1000] occurs 15 times, and thus, it is associated with the yellow color, and indicated correspondingly at the color scale at the right hand side. The same reasoning is adopted on Figure 8 where the maximum number of identical data points is 6.

Following this philosophy, we removed the color scale in Figures 13 and 14. In these two plots, the data point of [9 999,9 999] (indicated by yellow color) clearly dominates by 211 occurrences in both cases, whereas the rest of the points in the data cloud is represented by 1 to 6 occurrences, and therefore, the color scale has no added value.

3) Figs 7-8: Some of the circles appear as chopped-off circles, since they have a straight edge. Maybe this needs to be fixed?

Good point. The deformed circles appeared for unknown reason during the process of saving the displayed Matlab plots to files. The problem was fixed, when modifying the plots in relation with the previous comment.

4) Is the total number of entries in table 2 and table 3 different? Maybe write down the total, and explain the difference?

Yes, the total number of cases is different. To explain this fact, we modified the paragraph, by insertion of 2 sentences as follows (herein underlined).

‘The inter-comparison of the AWOS vs. LO data (RO vs. LO data) is presented in Table 2 (Table 3). Note that the total sum of cases is different in both tables: 181 (227) cases are reported in Table 2 (Table 3). The difference (46) stems in the inability of the standard AWOS to estimate the cloud cover on the basis of a single measurement of the ceilometer, and by means of the associated interpolation methods defined by the manufacturer.

These summaries reveal several characteristic features of the observation methods: […]’

5) Fig. 9: If it’s overcast at 12,000 ft, and if it’s scattered clouds at 6,000 ft, shouldn’t an all-sky monitor like LO or RO say that it’s overcast? Table 4 says that LO and RO say that it’s scattered clouds. Or are we only concerned about low clouds?

Yes, the clouds at 12,000 ft aren’t aeronautically significant, therefore, they are not reported in the standard aerodrome messages.

6) Table 10: These results for less than 600m are the only results from the extended validation experiment presented here. I realize that these short distances are more critical distances, but there are a lot more observations at mid-distances in Table 9, which might allow better statistics or better understanding.

We really investigated only 0-600 m category, as (1) it is aeronautically the most significant, and (2) it was of the main interest in the SESAR project, under which this research was performed.

7) The authors didn’t use a fisheye lens instead of the robotic camera with stitching. This choice can be motivated more.

The usual resolution of an outdoor camera chip is in order 6 Mpix. Using a fisheye camera implies that the WHOLE sky would be packed into a single image of a size of 6 Mpix, and the clouds would not be sufficiently sharp on a desktop monitor for the remote observers. With a rotating robotic camera, we have the chance to keep the resolution of the final composite even 1-2 orders higher (close to 61 x 6 Mpix).

To make our choice clearer, therefore, we appended the following sentences to Section 2.2:

‘Even though the procedure of image stitching is not trivial, we decided to use a rotating robotic camera instead of the commonly adopted fisheye lens approach [10-13]. While the latter methodology produces easy-to-get images, their resolution may not be sufficient for the remote observers, for instance to distinguish sharp edges of the cloud layers. ‘

8) Citations needed for the consensus stated on line 623.

We have smoothed the original sentence

‘There is a consensus in the scientific literature that observations of the prevailing visibility and cloud coverage by human observers is inherently biased by subjectivity.’

to the following one:

‘Since the observations of the prevailing visibility and cloud coverage are carried out by human observers, using the tool named ‘human eye’, and with no other objective measurement devices, the result is inherently biased by subjectivity.’

9) Complaining about lack of grant-proposal approval on line 650 is not the best thing to write in a journal article.

The original sentence

‘There was an attempt from the side of the authors of this article, to design and carry out such an experiment, but their proposal of the scientific research project has not yet been granted by the targeted grant agency.’

was replaced by the following one:

‘It is in the future plans of the authors of this article, to design and carry out such an experiment.’

10) Line 670: talking about what is underway at the time of manuscript preparation is not really the most appropriate thing to write about in a journal article. Maybe just refer to ‘future work’ or ‘ongoing work’, etc.

We agree with the Reviewer’s opinion. Therefore, the following sentence was removed from Line 670:

‘Such a study is under its way at the time of the manuscript preparation.’

and the remaining part was modified as follows:

‘A new, more elaborated analysis in the near future is expected to cover a period of length of about two years and will include a visual analysis of the camera records of several thousands of weather situations by remote observers.’

11) Line 676: do the authors mean refs. 11 & 12, instead of refs. 10 & 12?

It is obviously a typo, and we appreciate the fact that the Reviewer also verified the correctness of details like that. Corrected.

12) Line 694: maybe change ‘attempted to discuss’ to ‘discusses’ or ‘discussed’?

Corrected to ‘discusses’.

13) Line 753: maybe change ‘for independent research (paper)’ to ‘for a future paper’ or ‘for a companion paper’.

Corrected to ‘for a companion paper’.

Round 2

Reviewer 2 Report

I confirm that revised version addressed the concerns of the reviewers sufficiently so I suggest that it is published as is.

Author Response

We would like to thank to Reviewer #2 for his/her short and positive feedback. We are happy that our reply addressed all the open questions in the desired way.

Reviewer 3 Report

This paper submitted to the Applied Sciences journal (published by MDPI), by Bartok, Ivica, Gaál, Bartoková, and Kelemen, entitled ‘A novel camera-based approach to increase the quality, objectivity and efficiency of aeronautical meteorological observations’ is a well-written paper, and I am particularly impressed with its first revision after our reviews. Again, I think that the problem is an important one for aviation, and an interesting one, from an applied sciences point of view. The use of the English language and grammar is nearly perfect. The authors have mostly taken care of issues raised in my prior critique of the original manuscript.  In their response to my possible concerns and in their paper revisions, I no longer have any ethical concerns with their work and their presentation. The comparability of Local Observing with Remote Observing that is demonstrated for ‘minimum cloud base height by comparing Figure 8 with Figure 7’ and for ‘maximum cloud cover by comparing Table 3 with Table 2’ is quite impressive. As noted below, I only have 2 major observations (#A’ & #B’) and one observed typo (#C’) this time, which I think the authors should address prior to publication.

A’) There is no need to use gendered language in this paper (i.e., “manned” , “he”, “his”). I suggest that such language be changed or adapted.

B’) Referring to line 263 and to the authors’ nice response to my observation #A in my prior review, I would say that in the system’s results for “minimum cloud base height”, as described in this paper, only step#2 is presented for non-fog clouds. Steps #1 & #2 are presented here for fog clouds. And step#2 is skipped if there are no tall objects near the airport (though in this paper, there are tall objects nearby). In the case of skipping step#2, as in an airport without tall objects nearby, then only step#1 is used (out of steps #1 and #2). So for this paper and this system to be useful generally,  then the results of step #1 for non-fog clouds should be presented. By non-fog-clouds, I mean non-ground-hugging clouds that have cloud bases above the observing tower and the local hills. Maybe step#1 is possible for estimating the “minimum cloud base height” for both non-fog clouds and fog-clouds, but such an analysis is missing for non-fog clouds in this paper. I still think the authors need a discussion of how cloud height (say ‘above 1000 feet or so’) would be estimated by a Remote Observer without tall features nearby and without a ceilometer. Using language used in the new version (v2) of the paper, they need a discussion and perhaps an analysis for how a Remote Observer would estimate minimum cloud base height (above 1000 feet) using only step#1 and not step#2 nor step#3.

C’) Typo: Line 136: “this type of sensors,”  —> “this type of sensor,”

Author Response

Response to Reviewer #3

We would like to thank to Reviewer #3 for his positive feedback on the revised manuscript, and, again, for his constructive comments in the current, second round of revision.

Herein we respond step by step on the opened questions.

A’) There is no need to use gendered language in this paper (i.e., “manned”, “he”, “his”). I suggest that such language be changed or adapted.

Three occurrences of the term ‘manned’ were replaced by the word ‘human’ (two times in the abstract). Concerning the use of the pronouns ‘he’ and ‘his’, we tried to do our best to reach a gender-neutral formulation of our expressions in the manuscript (the occurrence of ‘his’/’he’ eliminated 2x/4x).

B’) Referring to line 263 and to the authors’ nice response to my observation #A in my prior review, I would say that in the system’s results for “minimum cloud base height”, as described in this paper, only step#2 is presented for non-fog clouds. Steps #1 & #2 are presented here for fog clouds. And step#2 is skipped if there are no tall objects near the airport (though in this paper, there are tall objects nearby). In the case of skipping step#2, as in an airport without tall objects nearby, then only step#1 is used (out of steps #1 and #2). So for this paper and this system to be useful generally, then the results of step #1 for non-fog clouds should be presented. By non-fog-clouds, I mean non-ground-hugging clouds that have cloud bases above the observing tower and the local hills. Maybe step#1 is possible for estimating the “minimum cloud base height” for both non-fog clouds and fog-clouds, but such an analysis is missing for non-fog clouds in this paper. I still think the authors need a discussion of how cloud height (say ‘above 1000 feet or so’) would be estimated by a Remote Observer without tall features nearby and without a ceilometer. Using language used in the new version (v2) of the paper, they need a discussion and perhaps an analysis for how a Remote Observer would estimate minimum cloud base height (above 1000 feet) using only step#1 and not step#2 nor step#3.

We understand the line of reasoning of the Reviewer. Nevertheless, an airport without a ceilometer is a rare case. The airports are aware of the fact that the ceilometer height measurement is a crucial parameter to be passed to pilots.

At less important non-airport meteorological stations, it is more common to have a human observer without a support of a ceilometer. But these categories of stations are not our target. They even have different, far less precise reporting rules that can be better fulfilled by step #1, i.e. by recognizing the cloud types and estimating their typical heights. But again, this is not our case. We target airports, and they tend to be the best equipped meteorological stations.

C’) Typo: Line 136: “this type of sensors,” —> “this type of sensor,”

Corrected.

Round 3

Reviewer 3 Report

This paper submitted to the Applied Sciences journal (published by MDPI), by Bartok, Ivica, Gaál, Bartoková, and Kelemen, entitled ‘A novel camera-based approach to increase the quality, objectivity and efficiency of aeronautical meteorological observations’ is a well-written paper, and I was particularly impressed with its first revision after our reviews; but the second revision after our reviews is still incomplete. Again, I think that the problem is an important one for aviation, and an interesting one, from an applied sciences point of view. The use of the English language and grammar is nearly perfect. The authors had mostly taken care of issues raised in my 1st critique of the original manuscript. They have taken into account my observation #A’ about gendered language in the 2nd version of the paper.  But I think they still need to add maybe a lengthy paragraph of discussion in the introduction and possibly also in section 2.2 of the paper regarding my observation #B’ about the 2nd version of their paper (which I quote below).  As noted below, I have 3 major observations (#A’’, #B’’, #C’’) this time, which I think the authors should address prior to publication.

A’’) Can the authors explain why they are referring to my 2nd review with the gendered pronoun ‘his’, in their authors’ response? They should not be guessing what my gender is, since this is an anonymous review.

B’) Referring to line 263 [in revision v2] and to the authors’ nice response to my observation #A in my [1st] review, I would say that in the system’s results for “minimum cloud base height”, as described in this paper, only step#2 is presented for non-fog clouds. Steps #1 & #2 are presented here for fog clouds. And step#2 is skipped if there are no tall objects near the airport (though in this paper, there are tall objects nearby). In the case of skipping step#2, as in an airport without tall objects nearby, then only step#1 is used (out of steps #1 and #2). So for this paper and this system to be useful generally,  then the results of step #1 for non-fog clouds should be presented. By non-fog-clouds, I mean non-ground-hugging clouds that have cloud bases above the observing tower and the local hills. Maybe step#1 is possible for estimating the “minimum cloud base height” for both non-fog clouds and fog-clouds, but such an analysis is missing for non-fog clouds in this paper. I still think the authors need a discussion of how cloud height (say ‘above 1000 feet or so’) would be estimated by a Remote Observer without tall features nearby and without a ceilometer. Using language used in the new version (v2) of the paper, they need a discussion and perhaps an analysis for how a Remote Observer would estimate minimum cloud base height (above 1000 feet) using only step#1 and not step#2 nor step#3.

B’’) The authors did respond to my observation #B’ in their authors’ response to review#2, but no action was taken in the paper itself. If the authors’ fall-back approach to estimating cloud-height (when step#2 = ‘their proposed technique’ is not available or applicable) is step#3 = ‘standard ceilometer measurements’, then this should be stated in the paper and discussed. The step#3 (ceilometer) is a (nearby) point measurement, whereas step#2 (their approach) is valid at the multiple (?) points (that are most-commonly) far away on/near the horizon where clouds might be low enough to cover tall objects. Maybe the ‘infrared-camera cloud-base-temperature measurements that the authors suggest (in passing)’ is therefore a preferable technique, instead of the RGB camera technique that the authors have implemented and tested? Then with the infrared-camera technique, there would be whole-sky coverage of the cloud base-height measurement, instead of just above the ceilometer or just above the tall objects on/near the horizon. But I am uncertain how accurate this infrared-camera technique would be across the whole sky and in daytime vs. nighttime.

C’’) Also, for the tall objects on/near the horizon, do they (i.e., the mountains) sometimes cause the clouds that are near them to have different-altitude cloud bases than the clouds nearer the airport? If so, how much does this effect cause biases in the authors’ measurement technique of prevailing cloud-base height?

Author Response

Response to Reviewer #3

We would like to thank, repeatedly, to Reviewer #3 for his/her feedback on the revised manuscript, in the third round of revision.

Here are the replies to the opened issues.

A’’) Can the authors explain why they are referring to my 2nd review with the gendered pronoun ‘his’, in their authors’ response? They should not be guessing what my gender is, since this is an anonymous review.

We apologize for this mistake, it was surely not intentional. Having focused on the scientific aspects of the reply, we just missed to double check the correct using of the pronouns his/her.

B’) Referring to line 263 [in revision v2] and to the authors’ nice response to my observation #A in my [1st] review, I would say that in the system’s results for “minimum cloud base height”, as described in this paper, only step#2 is presented for non-fog clouds. Steps #1 & #2 are presented here for fog clouds. And step#2 is skipped if there are no tall objects near the airport (though in this paper, there are tall objects nearby). In the case of skipping step#2, as in an airport without tall objects nearby, then only step#1 is used (out of steps #1 and #2). So for this paper and this system to be useful generally, then the results of step #1 for non-fog clouds should be presented. By non-fog-clouds, I mean non-ground-hugging clouds that have cloud bases above the observing tower and the local hills. Maybe step#1 is possible for estimating the “minimum cloud base height” for both non-fog clouds and fog-clouds, but such an analysis is missing for non-fog clouds in this paper. I still think the authors need a discussion of how cloud height (say ‘above 1000 feet or so’) would be estimated by a Remote Observer without tall features nearby and without a ceilometer. Using language used in the new version (v2) of the paper, they need a discussion and perhaps an analysis for how a Remote Observer would estimate minimum cloud base height (above 1000 feet) using only step#1 and not step#2 nor step#3.

In order to address the points of the Reviewer, we modified the manuscript in the following way.

After the bullet points listing Steps #1-3 in Section 2.2, the following parts were inserted:

The steps are not necessarily performed in the described order. The specific cases with no country features nearby (i.e., without Step #2) and/or with no ceilometer (i.e., without Step #3) will be discussed in Section 4 (Discussion).

In our system, the remote human observer can perform Steps #1 and #2 remotely by cameras and Step #3 by a remote access to the ceilometer data, which are displayed in the same system as the camera imagery. In this way, it is ensured that the remote observer has the ability to perform the same steps as the local observer.

The rest of the discussion was inserted into the Discussion Section:

There is another specific feature of the current analysis, and it is related to the fact that the target destination is surrounded by mountains (the High Tatras from North). Consequently, Steps #1-3 to determine the cloud base height (described in Section 2.2) can be performed with no limitations. In sites with no mountains or tall country features nearby, however, Step #2 has to be skipped. In such case, the LO is restricted to only follow Steps #1 and #3, and naturally, the same also holds true for the RO. This might lead to different results. Unfortunately, we have not got data from a different airport without mountains or tall features to verify the consequences of (the lack of) the particular settings. Nonetheless, we expect that both the local and the remote observers would be influenced in a similar way, since in the case of a non-mountainous airport, both LO and RO would suffer of the absence of Step #2.

A paragraph related to the lack of the ceilometer was also inserted (in our second reply to the Reviewer, it was only included in the reply, but not in the manuscript itself):

In principle, a very specific case would be an estimation of the cloud base height even without Step #3, i.e., with no ceilometer data. In fact, though, an airport without a ceilometer is a rarity. At less important, non-airport meteorological stations, it is more common to have a human observer without a support of a ceilometer. They even have different, far less precise reporting rules that can be better fulfilled by Step #1, i.e., by recognizing the cloud types and estimating their typical heights. This is, however, not our case. We target airports, and they tend to be the best equipped meteorological stations.

The Reviewer suggested to extend the Introduction and Section 2.2 with these ideas. Nevertheless, we found that the best position of these could be Section 2.2, and most importantly, the Discussion Section. First, we felt that these ideas would break the line of thoughts of the Introduction. Secondly, the new ideas are of a discussion character, and therefore, we are convinced that their right position is in the Discussion. The last reason is of a technical character. The new paragraphs refer to the nearby mountains (whereas the target site is only introduced later, in Section 2.4) and also use the abbreviations of LO and RO (and these are, again, defined later onward on the manuscript). We therefore hope that the Reviewer will understand that we did not follow his/her instructions strictly.

B’’) The authors did respond to my observation #B’ in their authors’ response to review#2, but no action was taken in the paper itself. If the authors’ fall-back approach to estimating cloud-height (when step#2 = ‘their proposed technique’ is not available or applicable) is step#3 = ‘standard ceilometer measurements’, then this should be stated in the paper and discussed. The step#3 (ceilometer) is a (nearby) point measurement, whereas step#2 (their approach) is valid at the multiple (?) points (that are most-commonly) far away on/near the horizon where clouds might be low enough to cover tall objects. Maybe the ‘infrared-camera cloud-base-temperature measurements that the authors suggest (in passing)’ is therefore a preferable technique, instead of the RGB camera technique that the authors have implemented and tested? Then with the infrared-camera technique, there would be whole-sky coverage of the cloud base-height measurement, instead of just above the ceilometer or just above the tall objects on/near the horizon. But I am uncertain how accurate this infrared-camera technique would be across the whole sky and in daytime vs. nighttime.

Using an infrared camera is a novel technique not only for RO, but also for LO. We are still in the phase of experiments with this technique. High quality infra cameras can have a precision of 1-2 degree Celsius, which corresponds to a vertical distance of 100-200 m in the atmosphere. But there are many variables and contaminations of the measurement to be solved, so this issue is not yet finished.

C’’) Also, for the tall objects on/near the horizon, do they (i.e., the mountains) sometimes cause the clouds that are near them to have different-altitude cloud bases than the clouds nearer the airport? If so, how much does this effect cause biases in the authors’ measurement technique of prevailing cloud-base height?

We addresses this issue in the Discussion section, by adding the following paragraph:

Note that both the LO and the RO should consider the fact that a cloud layer can have different height just above the airport and when it ‘touches’ the nearby mountains. However, the cloud observations in METAR should be representative of area within a radius of approximately 16 km of the aerodrome reference point [15], therefore, possible height differences over these 16 km are not taken into account.
